# The Implicit Delta Method

**Nathan Kallus**[*]
Cornell University & Netflix Research
kallus@cornell.edu

**James McInerney**[*]
Netflix Research
jmcinerney@netflix.com

## Abstract

Epistemic uncertainty quantification is a crucial part of drawing credible conclusions from predictive models, whether concerned about the prediction at a given point or any downstream evaluation that uses the model as input. When the predictive model is simple and its evaluation differentiable, this task is solved by the delta method, where we propagate the asymptotically-normal uncertainty in the predictive model through the evaluation to compute standard errors and Wald confidence intervals. However, this becomes difficult when the model and/or evaluation becomes more complex. Remedies include the bootstrap, but it can be computationally infeasible when training the model even once is costly. In this paper, we propose an alternative, the implicit delta method, which works by infinitesimally regularizing the training loss of the predictive model to automatically assess downstream uncertainty. We show that the change in the evaluation due to regularization is consistent for the asymptotic variance of the evaluation estimator, even when the infinitesimal change is approximated by a finite difference. This provides both a reliable quantification of uncertainty in terms of standard errors as well as permits the construction of calibrated confidence intervals. We discuss connections to other approaches to uncertainty quantification, both Bayesian and frequentist, and demonstrate our approach empirically.

## 1 Introduction

In this paper, we consider quantifying uncertainty in evaluations of predictive models trained on data. Consider the following examples. We fit a complex model (such as a neural net) to predict mean service time for an incoming call to a call center given some features, and we use it to prioritize calls in a queuing system. We may be interested in confidence intervals on the average wait time of incoming calls in the queue. Such confidence intervals would be crucial for drawing *credible* conclusions about such evaluations, since we know we cannot take the point prediction at face value given the sampling uncertainty in the data. We may, alternatively, be fitting a ranking algorithm by predicting user interaction from user-item features and then applying some fixed business rules on top, and we want to assess how often certain item categories would end up at the top. Of course, we would want to understand how certain we are in this assessment. Or, we fit a complex model to predict mean demand given price and user features from a price experiment we ran, and we use it to target discounts by optimizing demand at a price times unit profit. We may be interested in confidence intervals on the average profit over a given distribution of features.

All of these examples have three important features: they involve (1) a computationally burdensome step of fitting a large-scale model, (2) evaluating the result using a complicated function that need not even be known explicitly, and (3) requiring the epistemic uncertainty of the evaluation given a model and finite data set, in contrast to the total uncertainty comprising both epistemic and irreducible aleatoric uncertainty [9]. Were the first two of these simple (simple model and simple function thereof), we could just use the classic delta method [10] (see next section for detail). However, when these aspects are complex and the model involves many parameters, it may be too prohibitive to

---

[*]Equal contribution, alphabetical order.

36th Conference on Neural Information Processing Systems (NeurIPS 2022).

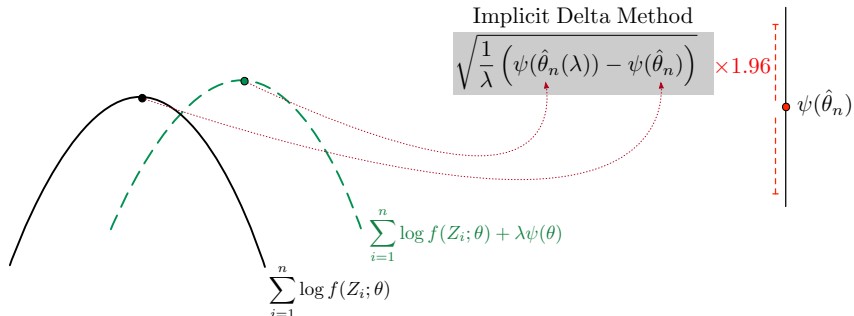

Figure 1: Illustration of how the implicit delta method (IDM) estimates 95% confidence intervals for a statistic of interest $\psi(\theta_0)$. Both the original MLE objective and the $\psi$-regularized objective are optimized, the $\psi$-evaluations of the two fitted models are compared, and $1.96$ multiples of the square root of the difference in evaluations is added and subtracted from the nominal evaluation to create a 95% confidence interval.

either analytically derive the whole inverse Fisher information matrix in the many model parameters or compute and invert the Hessian of the training loss empirically as well as compute the gradient of the final evaluation as a function of all parameters [28]. Even one aspect being complex may pose a serious challenge (*e.g.*, uncertainty quantification for the prediction of a complex model at a point). A remedy may be to bootstrap the whole process from data to final evaluation, but that can prove very computationally burdensome [12]. Usually just fitting the model once is already an expensive task; fitting it hundreds of times can be operationally infeasible. Other remedies, in the case of neural nets, may be the use of Langevin dynamics [39] or random dropout [15]. But these assess uncertainty in network weights and/or network predictions, which must then be translated to uncertainty in the final evaluation.

In this paper, we propose a direct yet inexpensive way to generically assess uncertainty in such settings. Specifically, we consider conducting inference when the estimator is some specified function of a (conditional) maximum likelihood estimator (MLE), such as a regression or classification model. Our proposal, the *implicit delta method*, works by simply adding an infinitesimal regularization to the MLE objective (*e.g.*, the sum of squared errors). We prove that the infinitesimal change in the final estimator due to this regularization is consistent for its asymptotic variance, the same variance that would have been predicted by the delta method in theory. Hence, the name of our method: we are conducting a delta-method quantification of uncertainty implicitly without explicitly propagating the uncertainty through the derivative of the evaluation function, analytically deriving the possibly-huge Fisher information matrix, or approximating it empirically. We prove that even when we approximate the infinitesimal change with a finite difference with constant width, the change we measure is still consistent for the asymptotic variance. This not only gives an assessment of uncertainty in terms of standard errors, it also permits us to construct calibrated confidence intervals. We demonstrate this in experiments, comparing to other popular approaches for uncertainty quantification, both Bayesian and frequentist.

## 2   Problem Set Up and the Delta Method

We consider an estimate constructed in two steps: first we fit a model using maximum likelihood estimation (MLE) and then apply some function to it. Namely, we consider data given by $n$ independent and identically distributed (iid) observations $Z_i \in \mathcal{Z}$, $i = 1, \ldots, n$, drawn from a population with density $f(z; \theta_0)$ with respect to some base measure $\mu$ on $\mathcal{Z}$. For example, the data may consist of observations of features $X$ and responses $Y$, with $Z = (X, Y)$.

In the first step, we fit a model to the data by MLE. Given a model $\{f(z; \theta) : \theta \in \Theta\}$ of densities (with respect to $\mu$) parametrized by $\theta \in \Theta \subseteq \mathbb{R}^d$, we set

$$\text{Model fitting:} \quad \hat{\theta}_n \in \operatorname*{argmax}_{\theta \in \Theta} \sum_{i=1}^{n} \log f(Z_i; \theta). \tag{1}$$

A prominent example is generalized regression, where we observe features and responses $Z = (X, Y)$, have a parametrized predictor $h_\theta(x) \in \mathbb{R}^p$, and a parametric model $g(y; \vartheta)$ with $\vartheta \in \mathbb{R}^p$. We then set

$f((x,y);\theta) = g(y;h_\theta(x))$.[2] Examples include least-squares regression, binary classification with cross-entropy loss, and Poisson regression, all with possibly complex and nonlinear predictors (*e.g.*, neural nets).

In the second step, we process the trained model in some way to come up with our estimate. Given some $\psi : \Theta \mapsto \mathbb{R}$, we compute

$$\text{Evaluation using fitted model:} \quad \hat{\psi}_n = \psi(\hat{\theta}_n).$$

One example in the case of generalized regression is evaluation of the predictor at a specified point, $\psi(\theta) = g_\theta(x_0)$. Another example is, when $g_\theta(x,p)$ corresponds to predicted mean demand at price $p$ given features $x$, we may be interested in the average optimal profit, $\psi(\theta) = \frac{1}{m}\sum_{j=1}^m \left(\sup_{p \geq c} g_\theta(x_j,p)(p-c)\right)$, for an evaluation dataset $\{x_j : j = 1,\ldots,m\}$. More generally, $\psi$ could be more opaque: it could involve, for example, simulating a queuing system with a controller parametrized by $\theta$, such as a priority policy with priority score $g_\theta(x)$.

We are interested in conducting uncertainty quantification for $\hat{\psi}_n$, and in particular in inference on its population limit, $\psi_0 = \psi(\theta_0)$. One way to do this inference is to propagate through $\psi$ the uncertainty within $\hat{\theta}_n$ about $\theta_0$, provided we understand the latter uncertainty. This is the so-called *delta method*.

To apply it, we must first understand the uncertainty in $\hat{\theta}_n$. Provided some regularity holds, this uncertainty can be characterized by the *curvature* of the objective function $\mathbb{P}\ell(\theta,\cdot)$ at $\theta = \theta_0$: if the curvature is sharp (resp., flat) then perturbing the objective and minimizing $\mathbb{P}_n\ell(\theta,\cdot)$ instead does not (resp., does) move the minimizer far away. This curvature is exactly the *Fisher information matrix*:

$$I(\theta) = -\int (\nabla^2 \log f(z;\theta))f(z;\theta)d\mu(z).$$

Specifically, under appropriate regularity conditions,

$$\sqrt{n}(\hat{\theta}_n - \theta_0) \rightsquigarrow \mathcal{N}(0, I^{-1}(\theta_0)). \tag{2}$$

In the above, $\mathcal{N}(\mu,\Sigma)$ refers to the multivariate normal distribution and $\rightsquigarrow$ refers to convergence in distribution. There are a variety of specific technical conditions that can establish this result. For an abstract presentation see theorems 9.27 and 9.28 in Wasserman [38]. For more rigorous treatments see theorem 13.2 of Wooldridge [41], theorem 3.3 of Newey and McFadden [27], theorem 5.1 of Lehmann and Casella [24], or theorem 8.3 of Davidson and MacKinnon [7], each of which uses slightly different technical regularity conditions.

Given Eq. (2) holds with $I(\theta_0) \succ 0$, the delta method would then guarantee that (see theorem 5.15 in [38])

$$\sqrt{n}(\hat{\psi}_n - \psi_0) \rightsquigarrow \mathcal{N}(0, V_0), \quad V_0 = \nabla\psi(\theta_0)^\top I^{-1}(\theta_0)\nabla\psi(\theta_0), \tag{3}$$

provided $\nabla\psi(\theta_0)$ exists and $V_0 > 0$.

An immediate and very important consequence of this is that we can construct calibrated confidence intervals for $\psi_0$: under Eq. (3),

$$\text{if} \quad n\hat{V}_n \to_p V_0, \quad \text{then} \quad \mathbb{P}\left(\psi_0 \in \left[\hat{\psi}_n \pm \Phi^{-1}((1+\beta)/2)\hat{V}_n^{1/2}\right]\right) \to \beta \quad \forall\beta \in (0,1). \tag{4}$$

where $\Phi$ refers to the cumulative distribution function of the standard normal distribution and $\to_p$ refers to convergence in probability. For example, as long as $I(\theta)$ and $\nabla\psi(\theta)$ are continuous at $\theta_0$, we can use

$$\hat{V}_n^{\text{DeltaMethod}} = \frac{1}{n}\nabla\psi(\hat{\theta}_n)^\top I^{-1}(\hat{\theta}_n)\nabla\psi(\hat{\theta}_n). \tag{5}$$

As discussed in the introduction, however, this approach may prove intractable in practice, especially when $\theta$ has many dimensions. Since we are only truly concerned with the uncertainty in $\hat{\psi}_n$ and not in $\hat{\theta}_n$, it may seem unnecessary and overly cumbersome to first compute the uncertainty in the latter and then propagate it. We next present our method, which does this all implicitly, never working directly with the vector $\theta$ except as an optimization variable in maximizing the MLE objective and a perturbation thereof.

---

[2]Note since we are not interested in the distribution of $X$ we here use only the conditional density of $Y \mid X$. Setting $f((x,y);\theta) = g(y;h_\theta(x))f(x)$ using the true unknown density $f(x)$ of $X$ does not change the MLE nor any of the results compared to omitting $f(x)$ altogether as we do here, which is referred to as the *conditional* MLE by Wooldridge [41].

# 3 The Implicit Delta Method

We would like to construct calibrated confidence intervals as in Eq. (4), but computing the estimated standard error as in Eq. (5) can be prohibitive. The IDM is a way to compute the estimated standard error while *neither* explicitly computing the uncertainty in $\hat{\theta}_n$ *nor* propagating this uncertainty through $\psi$. Instead, we will simply slightly perturb the original MLE in Eq. (1) using a little bit of regularization, which will implicitly do both of these difficult tasks for us.

To define the IDM, we first define a regularized version of the MLE. Given any $\lambda \geq 0$, we consider adding the regularizer $\lambda\psi(\theta)$ to Eq. (1) as well as the corresponding final estimator after passing through $\psi$:

$$\hat{\theta}_n(\lambda; \psi) \in \underset{\theta \in \Theta}{\operatorname{argmax}} \sum_{i=1}^{n} \log f(Z_i; \theta) + \lambda\psi(\theta), \quad \hat{\psi}_n(\lambda) = \psi(\hat{\theta}_n(\lambda; \psi)). \tag{6}$$

We refer to this as $\psi$-regularized MLE.

We then define the *infinitesimal* IDM (IIDM) as the infinitesimal change (*i.e.*, derivative) in our final estimate using $\psi$-regularized MLE as we infinitesimally increase $\lambda$ from 0:

$$\hat{V}_n^{\mathrm{IIDM}} = \left. \frac{\partial}{\partial \lambda} \hat{\psi}_n(\lambda) \right|_{\lambda=0} = \lim_{\lambda \to 0} \frac{1}{\lambda} \left( \hat{\psi}_n(\lambda) - \hat{\psi}_n \right). \tag{7}$$

Our first result shows that the IIDM estimate is consistent for the true asymptotic variance in Eq. (3).

**Theorem 1.** *Suppose that $\hat{\theta}_n \to_p \theta_0 \in \mathrm{Interior}(\Theta)$, $I(\theta_0) \succ 0$, and that, in a neighborhood of $\theta_0$, $\psi(\theta)$ is continuously differentiable and $f(Z; \theta)$ is almost surely twice continuously differentiable in $\theta$ with a Hessian that is bounded in operator norm by an integrable function of $Z$. Then*

$$n\hat{V}_n^{\mathrm{IIDM}} \to_p V_0.$$

The significance of Theorem 1 is that, per Eq. (4), it implies that $\left[ \hat{\psi}_n \pm \Phi^{-1}((1+\beta)/2) \sqrt{\hat{V}_n^{\mathrm{IIDM}}} \right]$ is a calibrated $\beta$-confidence interval for $\psi_0$.

Note that, aside from conditions on $\psi$ (which are the same as needed for Eqs. (3) and (5) to work), the regularity conditions required in Theorem 1 are implied by the regularity conditions required for establishing Eq. (2) by, for example, any of Davidson and MacKinnon [7], Lehmann and Casella [24], Newey and McFadden [27], Wooldridge [41]. In that sense, these conditions are not strong as they are already needed for $\hat{V}_n^{\mathrm{DeltaMethod}}$ to be a good estimate of uncertainty to begin with, and they fit into the existing framework for the asymptotic analysis of MLE.

The implication of Theorem 1 is that we may be able to implicitly complete the steps of the delta method (compute the uncertainty in $\hat{\theta}_n$, then propagate it through $\psi$) by simply assessing the impact of regularizing the MLE. However, this requires we actually differentiate with respect to the regularization coefficient. While this requires computing just one first derivative (rather than many first and second derivatives as in Eq. (5)), it is still not clear how to do this in practice.

In practice, we might approximate this derivative using finite differences, *i.e.*, replace the limit in Eq. (7) with a very small $\lambda$. This gives rise to what we call the finite-difference IDM (FDIDM), defined as follows for a given $\lambda_n > 0$:

$$\hat{V}_n^{\mathrm{FDIDM}} = \frac{1}{\lambda_n} \left( \hat{\psi}_n(\lambda_n) - \hat{\psi}_n \right). \tag{8}$$

Our next result shows that it in fact suffices to choose $\lambda_n$ constant. In fact any choice of $\lambda_n$ growing strictly slower than $n$, yields that $n\hat{V}_n^{\mathrm{FDIDM}}$ is also consistent for $V_0$, just like $\hat{V}_n^{\mathrm{IIDM}}$, provided just slightly more regularity holds.

**Theorem 2.** *Fix any $\lambda_n = o(n)$. Suppose that in addition to the assumptions of Theorem 1, in a neighborhood of $\theta_0$, $\psi(\theta)$ is thrice continuously differentiable and $f(Z; \theta)$ is almost surely thrice continuously differentiable in $\theta$ with a third-order derivative that is bounded in operator norm by an integrable function of $Z$. Then*

$$n\hat{V}_n^{\mathrm{FDIDM}} \to_p V_0.$$

It may seem surprising that a constant $\lambda_n$ suffices or that $\lambda_n$ is even allowed to grow, but that can be seen as an artifact of the fact we did not normalize the sum over the data in Eq. (6) by $1/n$. If we did normalize, it would be equivalent to rescaling $\lambda$ by $n$, so that $o(n)$ becomes $o(1)$, *i.e.*, requiring a vanishing increment for the finite differencing. Nonetheless, writing Eq. (6) as we did is very convenient, as it matches how one usually applies optimization algorithms such as stochastic gradient descent to training objectives, and it makes the choice of $\lambda_n$ for Eq. (8) very easy: just fix some constant and do not worry about the scaling with $n$. For example, setting $\lambda_n = 1$ suggests a very simple-looking 95%-confidence interval: $\left[\hat{\psi}_n \pm 1.96\sqrt{\hat{\psi}_n(1) - \hat{\psi}_n}\right]$. Note that it is not necessarily *better* to choose smaller $\lambda$: the smaller $\lambda$ the closer $\hat{V}_n^{\mathrm{FDIDM}}$ is to $\hat{V}_n^{\mathrm{IIDM}}$, but that need not mean it is a better estimate (see numerical illustration in Fig. 3). Finally, note that Eq. (8) is but one way to make a finite-difference approximation of a derivative, and other finite-difference formulae for derivatives (see ch. 4 of [4]) such as central differences could possibly be used.

**Remark 1** (Regression Using Squared Error Loss). When training regression models we usually minimize over model parameters (*e.g.*, neural net weights) the sum over the data of squared error loss, $\ell((x, y); \theta) = (y - g_\theta(x))^2$. This differs from the corresponding Gaussian log likelihood by a factor of $-\frac{1}{2\sigma^2}$ (and some constants that do not matter), where $\sigma^2$ is the residual variance of $Y$ given $X$. Therefore, to apply IDM, all we should do is simply regularize the sum-of-squared-errors *minimization* problem by $-2\sigma^2\lambda\psi(\theta)$, as that would be equivalent to dividing the log likelihood part by $-2\sigma^2$. Of course, we do not know $\sigma^2$, but we can estimate it by $\hat{\sigma}_n^2 = \frac{1}{n}\sum_{i=1}^n (y - g_{\hat{\theta}_n}(x))^2$, that is, the minimum average sum of squared errors. Since $\hat{\sigma}_n^2 \to_p \sigma^2$, as it is in fact the MLE estimate for $\sigma^2$, the asymptotic guarantees of Theorems 1 and 2 will continue to hold after this rescaling. Note that the standard errors given correspond to the MLE formulation of least-squares (usual standard errors) rather than the $M$-estimation formulation thereof (so-called robust or sandwich standard errors).

**Remark 2** (Using IDM to Compute the Fisher Information). A by-product of the proof of Theorem 1 is that, if we looked at the (vector-valued) derivative $\hat{W}_n = \left.\frac{\partial}{\partial\lambda}\hat{\theta}_n(\lambda; \psi)\right|_{\lambda=0} = \lim_{\lambda\to 0}\frac{1}{\lambda}\left(\hat{\theta}_n(\lambda; \psi) - \hat{\theta}_n\right)$, then $n\hat{W}_n \to_p I(\theta_0)^{-1}\nabla\psi(\theta_0)$. Therefore, if we set $\psi(\theta) = \theta_i$, *i.e.*, the $i^{\mathrm{th}}$ component of $\theta$, then $\hat{W}_n$ converges to the $i^{\mathrm{th}}$ column of $I(\theta_0)^{-1}$. Thus, by regularizing each component of $\theta$ in turn, we obtain the whole matrix.

Nonetheless, the whole raison d'être of IDM is to avoid working directly with the parameter vector $\theta$ altogether, and simply propagate its uncertainty automatically via the MLE optimization problem. For example, if we consider neural net regression, IDM would never make explicit reference to the vector of weights itself, only to the trained prediction model and its prediction performance on data. The above, wherein we compute the uncertainty in $\theta$ directly, stands in contradiction to this. Nonetheless, it can be a useful observation when inference on $\theta$ itself is for some reason of interest.

### 3.1 Extension to Multivariate Evaluations

We have so far focused on scalar evaluations for ease of presentation and as it covers the most important cases. We now show how our method easily extends to the multivariate case, where $\psi(\theta) = (\psi^{(1)}(\theta), \ldots, \psi^{(K)}(\theta)) \in \mathbb{R}^K$. The reason it may not suffice to run IDM separately for each component is that we may be interested in the *covariance* of the evaluations. Under the appropriate conditions, the extension of the delta method for MLE (Eq. (3)) to multivariate evaluations is

$$\sqrt{n}(\hat{\psi}_n - \psi_0) \rightsquigarrow \mathcal{N}(0, V_0), \quad V_0 = J(\theta_0)^\top I^{-1}(\theta_0)J(\theta_0), \tag{9}$$

where $J_{ij}(\theta) = \frac{\partial}{\partial\theta_i}\psi^{(j)}(\theta)$ is the $K \times d$ Jacobian of $\psi(\theta)$.

Our extensions of IIDM and FDIDM to multivariate evaluations are as follows:

$$\Delta_{ij}(\lambda) = \frac{1}{\lambda}\left(\psi^{(i)}(\hat{\theta}_n(\lambda; \psi^{(j)})) - \psi^{(i)}(\hat{\theta}_n)\right), \quad (\hat{V}_n^{\mathrm{IIDM}})_{ij} = \lim_{\lambda\to 0}\Delta_{ij}(\lambda), \quad (\hat{V}_n^{\mathrm{FDIDM}})_{ij} = \Delta_{ij}(\lambda_n).$$

**Theorem 3.** $n\hat{V}_n^{\mathrm{IIDM}} \to_p V_0$ *under the conditions of Theorem 1, and* $n\hat{V}_n^{\mathrm{FDIDM}} \to_p V_0$ *under the conditions of Theorem 2, both as $K \times K$ matrices.*

Surprisingly, this shows one need only solve $K + 1$ (possibly) regularized MLEs to get the full $K \times K$ covariance. (See Alg. 3 in supplement.)

## 3.2 Handling Non-differentiable Evaluations and Evaluation Uncertainty

So far we have assumed that the evaluation function is a known and differentiable function. Both statements may be false when we are interested in evaluating average performance on a population but we only have a finite evaluation data set and unit performance is not differentiable.

Specifically, let $W_1, \ldots, W_m \sim W$ denote the evaluation data set (which may be the same as the training set or otherwise dependent or it may be an independent data set) and let $h(w; \theta)$ the unit evaluation function. Consider the empirical evaluation map

$$\psi(\theta) = \frac{1}{m} \sum_{j=1}^{m} h(W_i; \theta).$$

If $h(W; \theta)$ is almost surely not differentiable in $\theta$, then $\psi$ is also almost surely not differentiable, which poses a challenge. We will show, however, that even though $\psi$ is not differentiable (which would break the usual delta method), FDIDM actually remains valid, *without any changes to the method*, provided certain on-average-differentiability holds.

To motivate the challenge of nondifferentiability and the plausibility of on-average-differentiability, consider an example where $g_\theta(x)$ represents an order quantity to stock in context $x$ and $w = (x, d)$ represents features and demand. If $h((x, d); \theta) = \max\{d - g_\theta(x), 0\}$ then $\psi(\theta)$ quantifies average unmet demand, but $h$ is not differentiable. Other non-differentiable examples include evaluating regression and classification models' performance using non-differentiable utility functions. While $h$ may not be differentiable and hence neither $\psi$, it may still be plausible that its expectation $\mathbb{E}[h(W; \theta)] = \mathbb{E}[\psi(\theta)]$ is differentiable. For example, if the distribution of demand conditioned on features is continuous, then the derivative of $\mathbb{E}[h(W; \theta)]$ in the example of average unmet demand will be the average of the conditional cumulative distribution function at $g_\theta(x)$ times $-\nabla_\theta g_\theta(x)$, and the second derivative will involve the conditional density.

We next show FDIDM *still* works with non-differentiable $\psi$, given some on-average-differentiability.

**Theorem 4.** *Consider $m = \Omega(n)$. Fix $\lambda_n = \lambda > 0$ constant. Suppose the assumptions of Theorem 2 hold, that Eq. (2) holds, that $h(W; \theta)$ is almost surely $L$-Lipschitz in $\theta$, and that for some $M > 0$,*

$$\lim_{\epsilon \to 0} \mathbb{P}\big(On \{\theta : \|\theta - \theta_0\| \le \epsilon\}, h(W; \theta) \text{ is twice differentiable in } \theta \text{ with } \|\nabla_\theta^2 h(W; \theta)\| \le M\big) = 1.$$

*Then*

$$(\hat{V}_n^{\mathrm{FDIDM}})^{-1/2}(\hat{\psi}_n - \psi_0) \rightsquigarrow \mathcal{N}(0, 1).$$

In the above example of average unmet demand, $L$ and $M$ would be bounds of the gradient and Hessian of $g_\theta(x)$ in $\theta$, and a sufficient condition for the assumption to hold would be that $g_\theta(x)$ is boundedly differentiable in $x$ for $\theta$ in a neighborhood of $\theta_0$ and $W = (X, D)$ has a continuous distribution.

Although the asymptotic variance of $\hat{\psi}_n - \psi_0$ is now different (in particular $\hat{V}_n^{\mathrm{DeltaMethod}}$ in Eq. (5) may be ill-defined), Theorem 4 shows that $\hat{V}_n^{\mathrm{FDIDM}}$ actually remains consistent for this new asymptotic variance. Thus, it provides a consistent estimate of standard errors and it still gives calibrated confidence intervals (note $\psi_0$ is now *random* but we can still have a confidence interval for it).

In some cases we may want to directly conduct inference on the population version of the evaluation, $\psi_0^* = \mathbb{E}[h(W; \theta_0)]$. To do this, all we have to do is simply also add the uncertainty due to finite evaluation data set. Under standard regularity conditions, we have

$$(\hat{V}_m^\psi)^{-1/2}(\psi_0 - \psi_0^*) \rightsquigarrow \mathcal{N}(0, 1), \quad \text{where} \quad \hat{V}_m^\psi = \frac{1}{(m-1)m} \sum_{j=1}^{m} (h(w; \hat{\theta}_n) - \psi(\hat{\theta}_n))^2.$$

Therefore, provided the training and evaluation data sets are independent,

$$(\hat{V}_n^{\mathrm{FDIDM}} + \hat{V}_m^\psi)^{-1/2}(\psi(\hat{\theta}_n) - \psi_0^*) \rightsquigarrow \mathcal{N}(0, 1).$$

If not independent, then $\sqrt{\hat{V}_n^{\mathrm{FDIDM}}} + \sqrt{\hat{V}_m^\psi}$ provides a consistent upper bound on the standard error.

## 3.3 Implementation

FDIDM is given in pseudocode in Alg. 1. Given an objective function $\mathcal{L} := \sum_i^n \log f(Z_i; \theta)$, evaluation function $\psi$, and scalar width $\lambda$, FDIDM returns the estimated variance of $\psi(\hat{\theta}_n)$. The first step is to maximize the original objective w.r.t. $\theta$. Usually, this task has already been solved as this is

---

**Algorithm 1:** Finite-difference implicit delta method (FDIDM)

---
**Input:** Learning objective $\mathcal{L}$, evaluation $\psi$, scalar $\lambda$

**1 Function** FDIDM($\mathcal{L}$, $\psi$, $\lambda$):

**2**    $\hat{\theta}_n \leftarrow \arg\max_\theta \mathcal{L}(\theta)$            `// optimize learning objective`

**3**    $\hat{\theta}_n(\lambda) \leftarrow \arg\max_\theta \mathcal{L}(\theta) + \lambda\psi(\theta)$      `// optimize `$\psi$`-regularized objective`

**4**    **return** $\frac{1}{\lambda}(\psi(\hat{\theta}_n(\lambda)) - \psi(\hat{\theta}_n))$      `// estimated variance of `$\psi(\hat{\theta}_n)$

---

the trained predictive model. Then, maximize the $\psi$-regularized objective w.r.t. $\theta$. Finally, return the estimated variance using the finite-difference method evaluated at $\lambda = 0$. See Appendix D for the corresponding algorithm when $\psi$ is multivariate.

In practice, one can further reduce the computational cost of FDIDM due to the fact that the $\psi$-regularized objective can be made arbitrarily close to original objective by choosing $\lambda$ small enough, subject to numerical instability at extremely small values. Specifically, when using stochastic gradient ascent in FDIDM, once the optimum $\hat{\theta}_n$ has been found, only a small number of gradient updates may be required to also find $\hat{\theta}_n(\lambda)$.

FDIDM also admits non-gradient-based approaches. Consider the case that $\psi$ is a simulator that takes a fitted model and returns a set of evaluations and no gradient. Then the $\psi$-regularized objective may optimized by gradient-free methods such as Nelder-Mead [26] and Bayesian optimization [14].

## 4 Alternatives for Uncertainty Quantification and Related Work

Uncertainty quantification in machine learning is a topic of major interest due to the need to make downstream inferences and decisions based on the predictions of large-scale networks trained on massive datasets in either a frequentist [16, 28, 31] or Bayesian fashion [3, 8, 15]. Our focus here is on methods that can flexibly isolate epistemic uncertainty in an evaluation, representing data sampling uncertainty of that evaluation under a given model. In cases where the total uncertainty for predictions is desired, a broader set of methods may be brought to bear, such as conformal prediction [1, 36, 37], Platt scaling [17, 33], or indeed, any of the aforementioned statistical methods used in conjunction with a term or terms for aleatoric uncertainty. An exhaustive account of the literature is outside the scope of this paper. We highlight the principal ideas and points of contact with our work.

**The Bootstrap** The bootstrap simulates sampling from the true data generating distribution by resampling from the observed dataset (see, *e.g.*, [12] for an introduction and [22] for theory on when it works). The key advantage is that it enables general-purpose and easy-to-implement uncertainty quantification for estimators. It comes at a high computational burden because the estimator, which may comprise a model-fitting algorithm and prediction, needs to be executed many times. In the context of deep learning, many useful adaptations of the bootstrap and the related jackknife have been proposed to increase its computational efficiency [16, 30, 31]. Maintaining an ensemble of models as a representation of the variability of the evaluation is an appealing intuition that does not restrict one to local approximations, and may be combined with local approximations where necessary.

**The Functional Delta Method** The delta method [10] is a classic approach that is widely used with small models with an analytic Fisher information matrix (*e.g.*, linear regression) and, more recently, auto-differentiation unlocks the delta method for a larger class of models [28]. The bottleneck is the need to calculate then invert the Fisher information matrix, for which there are various approximations [25, 32]. The delta method applies to a wide range of (differentiable) estimators subject to regularity conditions that ensure asymptotic normality of the parameter estimates and this constraint carries over to the implicit delta method. The functional delta method extends the delta method to evaluations of infinite-dimensional parameters (see Ch. 12 [22]) but is usually restricted to analytically deriving influence functions in theory by differentiating the population estimand with respect to distributions and then approximating the influence function by plugging in estimates of unknown nuisances [5, 18].

**Bayesian Uncertainty Quantification** Tractable methods for approximate Bayesian inference in neural networks, such as variational inference in feed-forward nets [3], autoencoders [21, 35], normalizing flows [34], dropout uncertainty [15], stochastic gradient Langevin dynamics [39] and related approaches, present an impressive range of options for uncertainty quantification. In cases

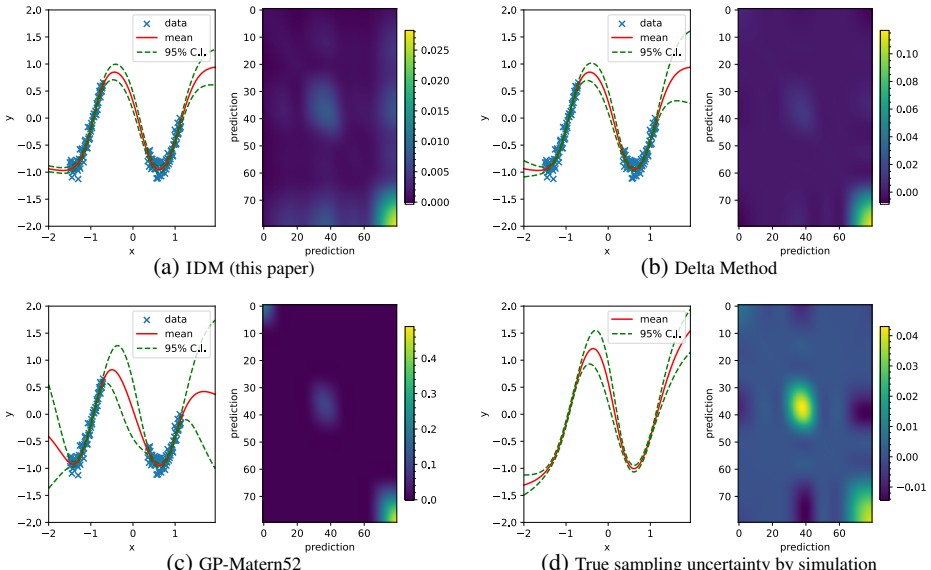

Figure 2: Fits along with uncertainty bounds and estimated prediction-covariance matrix for data generated from $y = -\sin(3x - \frac{3}{10}) + \frac{1}{10}\epsilon$, where $\epsilon \sim \mathcal{N}(0, 1)$

where it is sufficient to only consider a single mode in the posterior, local methods can prove useful. In particular, an alternative interpretation of the delta method is as a special case of the Laplace approximation to Bayesian inference, where Eq. (5) arises in the posterior predictive distribution for a local multivariate Gaussian approximation around the maximum *a posteriori* estimate. Several recent works have investigated the potential of the Laplace approximation as a way to avoid having to characterizing the full posterior in deep networks [8, 19, 20]. IDM can provide another way to perform a Laplace approximation and may be orthogonally combined with the above methods.

## 5 Experiments

In this section, we evaluate finite-difference implicit delta method (FDIDM) on a range of tasks that require confidence intervals.[3] Our goal is to quantify the extent to which FDIDM applies in practice and how it compares to alternative methods. We start with 1D synthetic data in Sec. 5.1 where we apply a neural net to recover known functions from small datasets. Then, in Sec. 5.2, we consider the task of inferring average utility under a neural net trained on a set of real-world benchmark datasets. In Sec. 5.3, we apply FDIDM to variational autoencoders and use the implicit delta perspective to understand the effect of KL down-weighting. We find that the motivation and convergence properties of FDIDM are empirically observed and this may be useful to practitioners seeking to quantify the epistemic uncertainty of complex models on a variety of regression and classification tasks.

### 5.1 1D Synthetic Examples

We consider known quadratic and sinusoidal functions from which we draw a random dataset. Fig. 2 gives the data generating stochastic function for a sin wave and the resulting fits for FDIDM, the classic delta method, a Gaussian process (GP) with Matern-52 kernel, as well as simulation from the true data generating function. (Appendices B and C provide the results on the quadratic function and further experimental details, respectively.) The quadratic example has evenly dispersed input data and there is close alignment between the methods. The sin wave is more challenging because it requires extrapolation – also known as "in-between" uncertainty in [13] – from outside the ranges of given inputs. Results for IDM, DM, and simulation are all based on estimates using a neural net with 1 hidden layer of 50 `tanh` units.[4] It should be noted that the GP is not trying to estimate the frequentist sampling variance (shown in the simulation results) but rather the Bayesian posterior uncertainty (although they can coincide asymptotically; [38, theorem 11.5]); we include it largely for a qualitative comparison to a popular epistemic-uncertainty quantification method. In particular, unlike

---

[3]The source code is available at `https://github.com/jamesmcinerney/implicit-delta`.

[4]This architecture is in line with [13], which also provided the basis for our sin example.

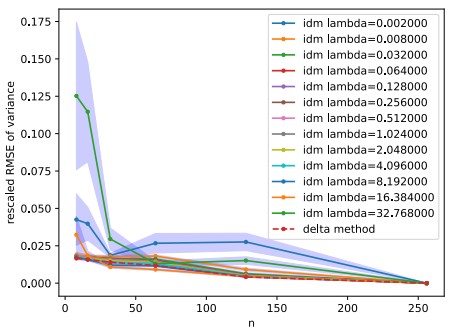

(a) Number of data points as independent variable.

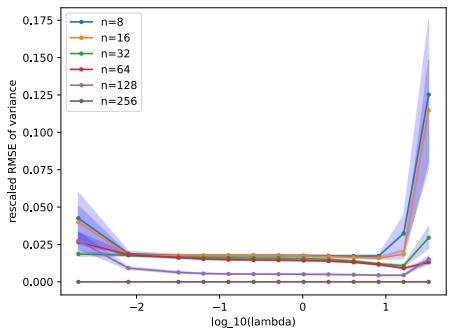

(b) $\lambda$ as independent variable.

Figure 3: Convergence of IDM in the quadratic task for different values of $n$ and $\lambda$. There is a wide dynamic range of acceptable $\lambda$.

Table 1: Run time (seconds)

|           | Vehicle | Waveform | Satellite | MNIST |
|-----------|---------|----------|-----------|-------|
| IDM       | 39      | 129      | 111       | 303   |
| Bootstrap | 806     | 2,334    | 3,192     | 7,164 |

the IDM, DM, and simulation results, the GP does not yield an interval around the neural-net based mean estimate and instead has a different mean function[5].

As expected, IDM agrees most with the delta method while the GP overestimates uncertainty, particularly for extrapolation at the outer edges. The corresponding full covariance matrix of the predictions is also given in Fig. 2. All the methods recover the high-level structure of covariance for both examples, though the scale factors differ considerably.

Fig. 3 shows the convergence of the root mean squared error of IDM w.r.t. the true variance as determined by 50 resamples from the data generating distribution in the quadratic task. The squared errors are rescaled by $n^2$ to account for decreasing scale ($\frac{1}{n}$) of the true the variance as $n$ grows. Shaded error bars indicate one standard error. Convergence for the standard delta method is also shown for reference. We find there is a wide dynamic range of acceptable values of $\lambda$. Small values of $\lambda < 0.01$ perform poorly, likely due to numerical instability, but performance improves for larger $\lambda$. The setting $\lambda = 0.512$ even outperforms the delta method. These findings support the implication of Theorem 2 indicating that convergence holds as long as $\lambda$ grows sublinearly to $n$.

## 5.2 Confidence in Predicted Cost Downstream of Classification

A set of classification tasks are fitted with a neural net with one hidden layer and 50 `tanh` hidden units. In this setting, we wish to calculate confidence intervals over total cost in a downstream task under predictions from the network. An arbitrary cost function is set up, in this case, the average cross entropy of the observations on a held-out validation dataset, though in practice we could have a wide variety of cost functions relating to the task downstream of the classifier. It is challenging to form a confidence interval for even this simple cost function because it is a function of predictions from the network. Under this scenario, it is typical to make a bootstrapping estimate, requiring $B$ times the cost of training the network (here, we use $B = 50$). FDIDM is also applicable in this setting. We show both methods on MNIST image classification [23] and a set of UCI benchmark datasets [11] in Fig. 4. We find that FDIDM has good coverage for a fraction of the computational cost of the bootstrap estimate. Specifically, a time complexity comparison is provided in Table 1.[6]

## 5.3 Down-Weighting KL in Variational Autoencoders

The variational autoencoder (VAE) is a prominent example of approximate inference in deep generative models [21, 35]. In practice, it has been observed that down-weighting the KL term in the variational objective by a factor $\frac{1}{T}$, where $T > 1$, results in significantly better accuracy on held-out

---

[5]We also applied the `arccos` kernel in a GP which imitates a neural network [6] but found that the Matern-52 kernel inferred a mean that was closer in practice to the mean inferred by the neural net.

[6]Run time was measured on a MacBook Pro 2.3 GHz Quad-Core Intel Core i7 with 32 GB RAM.

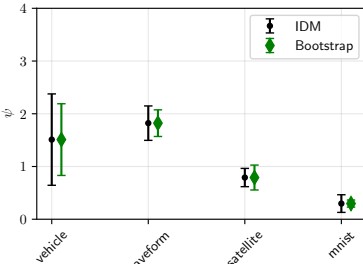
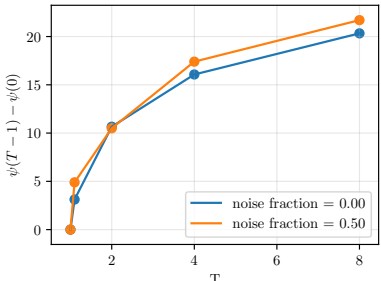

Figure 4: Predicted distribution of utility in classification on benchmark datasets. Datasets are shown in order of number of examples.

Figure 5: Improvement in reconstruction quality over the unweighted likelihood as a function of $T$.

test data [2, 42]. This is closely related to other ways of reweighting the prior and likelihood terms in approximate inference such as data augmentation [29] and the cold posterior effect [40]. Various explanations for the benefit of setting $T > 1$ have been posited, *e.g.*, model misspecification. Here, we briefly explore an IDM interpretation of this phenomenon.

Observing that any objective does not change its critical points under (non-zero) rescaling, it holds that down-weighting the KL term yields the same optimization problem as up-weighting the reconstruction error by $T$. We consider this setting and define $\psi$ to be the reconstruction quality (*i.e.*, negative reconstruction error) while optimizing the standard (unweighted) evidence lower bound defined by the VAE. Rearranging the terms in Eq. (8), letting $\lambda_n := T - 1$, we have

$$\hat{\psi}_n(\lambda_n) = (T - 1)\hat{V}_n^{\text{FDIDM}} + \hat{\psi}_n(0). \tag{10}$$

The two immediate implications of Eq. (10) for fixed $T > 1$ of $o(n)$ are that,

1. the reconstruction quality for the objective implied by variational inference is upper bounded by that of the objective with up-weighted likelihood; and,
2. the higher the variance in the reconstruction quality, as determined by the dataset and model, the greater the benefit of up-weighting the likelihood term in variational inference.

Since (1) is already supported by existing empirical work, we focus on evaluating whether (2) also holds in practice. To do this, we artificially increase the variance of the reconstruction quality by perturbing a proportion of the dataset and compare $\hat{\psi}_n$ for various $T$. Fig. 5 shows the results for applying a VAE to the MNIST dataset after adding i.i.d. noise $\epsilon \sim \text{Uniform}(0, \frac{1}{20})$ to each pixel in a randomly chosen fraction of the images. The figure indicates that, as predicted by Eq. (10), the gap in the reconstruction quality for $T > 1$ relative to $T = 1$ increases as more variance is introduced. Higher values of $T$ do indeed result in better reconstruction quality and this advantage grows with the amount of variance. In sum, these findings are consistent with the hypothesis that the advantage from KL down-weighting may be explained as the residual between the $\psi$-regularized variational objective and the objective implied by the evidence lower bound, though further studies are required.

## 6 Discussion, Limitation, and Conclusions

In this paper we develop the implicit delta method for forming calibrated confidence intervals via a careful regularization of the model objective. Like the delta method, the method requires certain regularity conditions (Theorem 1) and for the MLE to be at a stable optimum, where perturbations around the optimum reliably capture sampling uncertainty. If this is not the case, *e.g.*, the parameter has failed to converge or the objective itself is changing, it yields unreliable results. For this reason, IDM – like the delta method and the bootstrap – may be misleading for small data, and indeed uncertainty quantification with small data is fundamentally difficult. The most appealing feature of IDM is that it does not require the variance of the parameters to be made explicit, which also suggests future research in exploring the compatibility of nonparametric models with IDM. There is also the potential to explore IDM in constrained MLE and in general $M$-estimation.

## Acknowledgments

We are grateful for the insightful comments of the anonymous reviewers and our colleagues at Netflix.

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
