# Supplementary Materials

## Outline of Supplementary Materials

- Proofs of results stated in the main text are provided in Appendix A.
- Additional experimental results, including coverage plots, are provided in Appendix B.
- Additional details for all experiments, including specifics on the implementation of the optimization, are provided in Appendix C.
- The finite-difference IDM algorithm for multivariate-valued evaluations are detailed in Appendix D.

## A  Omitted Proofs

### A.1  Proof of Theorem 1

*Proof.* Let $\theta_0 \in \mathcal{N} \subset \Theta$ be the interior neighborhood where the assumed differentiable hold. Since $\hat{\theta}_n \to_p \theta_0$, we have that $\mathbb{P}\left(\hat{\theta}_n \in \mathcal{N}\right) \to 1$.

Define $L_n(\theta) = \sum_{i=1}^n \log f(Z_i; \theta)$ on $\mathcal{N}$. By assumptions of continuity and existence of envelope, lemma 4.3 of Newey and McFadden [4] guarantees that $-\frac{1}{n}\nabla^2 L_n(\hat{\theta}_n) \to_p I(\theta_0)$. Since $I(\theta_0) \succ 0$, we have by continuous mapping that $\mathbb{P}\left(\nabla^2 L_n(\hat{\theta}_n) \prec 0\right) \to 1$.

Consider the event $\hat{\theta}_n \in \mathcal{N}$ and $\nabla^2 L_n(\hat{\theta}_n) \prec 0$, which has probability tending to 1. Define $\mathcal{L}(\lambda, \theta) = \nabla L_n(\theta) + \lambda \nabla \psi(\theta)$ on $\mathbb{R} \times \mathcal{N}$. Note that $\mathcal{L}(0, \hat{\theta}_n) = 0$ and that the Jacobian of $\mathcal{L}$ with respect to $\theta$ at $(0, \hat{\theta}_n)$ is $\nabla^2 L_n(\hat{\theta}_n) \prec 0$. Therefore, there exists $\tilde{\theta}_n(\lambda)$ with $\mathcal{L}(\lambda, \tilde{\theta}_n(\lambda)) = 0$ and $\frac{\partial \tilde{\theta}_n}{\partial \lambda}(\lambda) = -(\nabla^2 L_n(\hat{\theta}_n(\lambda; \psi)))^{-1}\nabla \psi(\hat{\theta}_n(\lambda; \psi))$ for $\lambda \in (-u, u)$ for some $u > 0$. Moreover, since $\nabla^2 L_n(\hat{\theta}_n) \prec 0$ and by continuous derivatives, there is a $0 < u' \le u$ such that $\tilde{\theta}_n(\lambda)$ uniquely solves $\mathcal{L}(\lambda, \theta) = 0$ for $\lambda \in (-u', u')$ and therefore $\tilde{\theta}_n(\lambda) = \hat{\theta}_n(\lambda; \psi)$ for such $\lambda$. Finally, chain rule applied to $\psi(\hat{\theta}_n(\lambda; \psi))$ for $\lambda \in (-u', u')$ gives $\frac{\partial \hat{\psi}_n}{\partial \lambda}(\lambda) = \nabla \psi(\hat{\theta}_n(\lambda; \psi))^\top \frac{\partial \hat{\theta}_n}{\partial \lambda}(\lambda)$.

Applying this at $\lambda = 0$, we conclude that with probability tending to 1,

$$n\hat{V}_n^{\mathrm{IIDM}} = \nabla \psi(\hat{\theta}_n)^\top \left(-\frac{1}{n}\sum_{i=1}^n \nabla^2 \log f(Z_i; \hat{\theta}_n)\right)^{-1} \nabla \psi(\hat{\theta}_n).$$

By continuity, $\nabla \psi(\hat{\theta}_n) \to_p \nabla \psi(\theta_0)$. We have also already shown that $-\frac{1}{n}\nabla^2 L_n(\hat{\theta}_n) \to_p I(\theta_0) \succ 0$. Therefore, the proof is completed by continuous mapping. $\qquad \square$

### A.2  Proof of Theorem 2

*Proof.* The assumptions imply that $\hat{\psi}_n''(0)$ exists and $n^2 \hat{\psi}_n''(0)$ converges to a constant in probability. Applying Taylor's theorem using the Lagrange form of the remainder, we have that, for some random $\tilde{\lambda}_n \in [0, \lambda_n]$,

$$\hat{\psi}_n(\lambda_n) = \hat{\psi}_n(0) + \hat{\psi}_n'(0)\lambda_n + \hat{\psi}_n''(0)\tilde{\lambda}_n^2.$$

Rearranging yields

$$n\hat{V}_n^{\mathrm{FDIDM}} = n\hat{V}_n^{\mathrm{IIDM}} + n\hat{\psi}_n''(0)\tilde{\lambda}_n^2/\lambda_n.$$

By Theorem 1, $n\hat{V}_n^{\mathrm{IIDM}} \to_p V_0$.

Considering the remainder term, we have

$$\left| n\hat{\psi}_n''(0)\tilde{\lambda}_n^2/\lambda_n \right| \le \frac{\lambda_n}{n}\left| n^2 \hat{\psi}_n''(0) \right| = o(1)O_p(1) = o_p(1),$$

completing the proof. $\qquad \square$

### A.3 Proof of Theorem 3

*Proof.* From the proof of Eq. (7), we know that, with probability tending to 1, $\frac{\partial}{\partial\lambda}\hat{\theta}_n(\lambda;\psi^{(j)}) = -(\nabla^2 L_n(\hat{\theta}_n(\lambda;\psi^{(j)})))^{-1}\nabla\psi^{(j)}(\hat{\theta}_n(\lambda;\psi^{(j)}))$ (from the implicit function argument) and $\frac{\partial}{\partial\lambda}\psi^{(i)}(\hat{\theta}_n(\lambda;\psi^{(j)})) = \nabla\psi^{(i)}(\hat{\theta}_n(\lambda;\psi^{(j)}))^\top \frac{\partial}{\partial\lambda}\hat{\theta}_n(\lambda;\psi^{(j)})$ (from chain rule argument). Combining and applying for each entry $i, j$ yields the first statement. The second statement follows exactly as in Theorem 2 by a Taylor expansion. $\qquad\square$

### A.4 Proof of Theorem 4

*Proof.* Define $\dot{h}(W;\theta)$ as $\nabla h(W;\theta)$ when it exists and 0 otherwise. Define $\psi^*(\theta) = \mathbb{E}[h(W;\theta)]$. Further define

$p(\epsilon) = 1-\mathbb{P}\big(\text{On } \{\theta : \|\theta - \theta_0\| \leq \epsilon\}, h(W;\theta) \text{ is twice differentiable in } \theta \text{ with } \|\nabla^2_\theta h(W;\theta)\| \leq L\big),$

so that by assumption $p(\epsilon) = o(1)$. By continuity of probability, $p(0) = 0$ so that $h(W;\theta)$ is almost surely twice boundlessly differentiable at $\theta = \theta_0$. Thus, $\nabla\psi^*(\theta_0) = \mathbb{E}\dot{h}(W;\theta_0)$ exists.

Consider any sequence $\theta \to \theta_0$. Then, first by triangle inequality and then by Taylor's theorem,

$$\mathbb{E}\big[\big|\psi(\theta) - \psi(\theta_0) - \nabla\psi^*(\theta_0)^\top(\theta - \theta_0)\big|\big] \leq \mathbb{E}\Big[\big|h(W;\theta) - h(W;\theta_0) - \dot{h}(W;\theta_0)^\top(\theta - \theta_0)\big|\Big]$$

$$\leq p(\|\theta - \theta_0\|)L\|\theta - \theta_0\| + \frac{1}{2}M\|\theta - \theta_0\|^2$$

$$= o(\|\theta - \theta_0\|).$$

Since $\|\theta - \theta_0\| = O_p(1/\sqrt{n})$,

$$\psi(\hat{\theta}_n) = \psi(\theta_0) + \nabla\psi^*(\theta_0)^\top(\hat{\theta}_n - \theta_0) + o_p(1/\sqrt{n}).$$

Therefore, by Slutsky's theorem,

$$\sqrt{n}(\psi(\hat{\theta}_n) - \psi(\theta_0)) \rightsquigarrow \mathcal{N}(0, \nabla\psi^*(\theta_0)^\top I^{-1}\nabla\psi^*(\theta_0)).$$

Then if, hypothetically, we apply FDIDM with the unknown $\psi^*(\theta)$ to obtain $\tilde{V}_n^{\text{FDIDM}} = (\psi^*(\hat{\theta}_n(\lambda;\psi^*)) - \psi^*(\hat{\theta}_n))/\lambda$, then, by Theorem 2, we have that

$$\frac{\psi(\hat{\theta}_n) - \psi(\theta_0)}{\sqrt{\tilde{V}_n^{\text{FDIDM}}}} \rightsquigarrow \mathcal{N}(0, 1).$$

To complete the proof, we next argue that $\hat{V}_n^{\text{FDIDM}} - \tilde{V}_n^{\text{FDIDM}} \to_p 0$, that is, the difference vanishes between doing hypothetical FDIDM with the unknown $\psi^*$ and actual FDIDM with the known $\psi$. Let $\theta_0 \in \mathcal{N} \subset \text{Interior}(\Theta)$ denote the interior neighborhood where all desirable differentiabilities hold. Our assumptions yield $\mathbb{P}(\hat{\theta}_n \in \mathcal{N}, \hat{\theta}_n(\lambda;\psi) \in \mathcal{N}, \hat{\theta}_n(\lambda;\psi^*) \in \mathcal{N}) \to 1$ and $\sup_{\theta\in\mathcal{N}}|\psi(\theta) - \psi^*(\theta)| \to_p 0$ per lemma 4.3 of Newey and McFadden [4]. Then, on the event that $\hat{\theta}_n \in \mathcal{N}, \hat{\theta}_n(\lambda;\psi) \in \mathcal{N}, \hat{\theta}_n(\lambda;\psi^*) \in \mathcal{N}$, we have

$$\Big|\hat{V}_n^{\text{FDIDM}} - \tilde{V}_n^{\text{FDIDM}}\Big| = \lambda^{-1}\Big|\psi(\hat{\theta}_n(\lambda;\psi)) - \psi(\hat{\theta}_n) - \psi^*(\hat{\theta}_n(\lambda;\psi^*)) + \psi^*(\hat{\theta}_n)\Big|$$

$$\leq 2\lambda^{-1}\sup_{\theta\in\mathcal{N}}|\psi(\theta) - \psi^*(\theta)|,$$

yielding the statement. $\qquad\square$

## B  Further Experiments

Fig. 6 shows the fitted mean and covariance on a single draw of the quadratic dataset. With the inputs more dispersed (than the sin example) we see that all methods are in closer agreement. Fig. 7 gives the 95% coverage of the methods as a function of input. The quadratic function has close to 95% coverage across the inputs for all methods. The sin example restricts the set of available inputs (even on resample) and uses a neural network on a small dataset (N=160), both of which make it particularly challenging for coverage. There is some agreement among the methods that use the neural network (IDM and delta method). The GP-Matern52 comprises a different model of the mean and covariance and is included for reference. All of the methods are at least 20 percentage points away from the true coverage across the inputs.

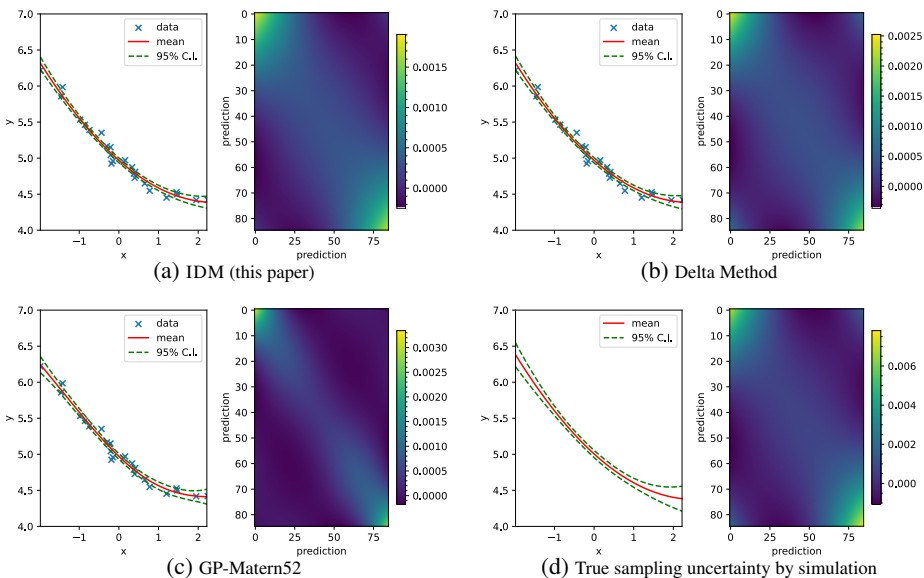

(a) IDM (this paper)

(b) Delta Method

(c) GP-Matern52

(d) True sampling uncertainty by simulation

Figure 6: Fits along with uncertainty bounds and estimated prediction-covariance matrix for data generated from $y = \frac{1}{10}x^2 - \frac{1}{2}x + 5 + \frac{1}{10}\epsilon$, where $\epsilon \sim \mathcal{N}(0, 1)$

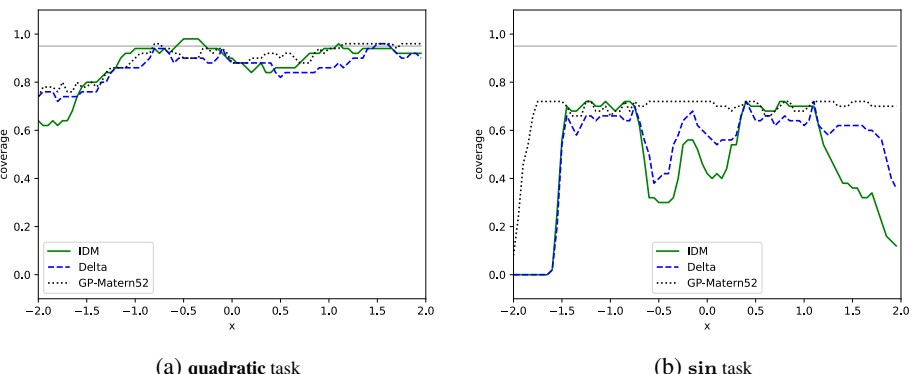

(a) **quadratic** task

(b) **sin** task

Figure 7: Coverage of 95% confidence intervals pointwise for prediction at each input.

## C   Details for Experiments

In this section we provide details for the experimental setup used in the paper.

**Stochastic Gradient Finite-Difference IDM**   The algorithm we propose for combining stochastic gradient ascent with finite-difference IDM is given in Alg. 2. This algorithm is used in all experiments. It estimates $\beta$-confidence intervals given a stochastic objective $\tilde{\mathcal{L}}$ that is an unbiased estimate of the true objective $\mathbb{E}[\tilde{\mathcal{L}}] = \mathcal{L}$ and evaluated each iteration $i$ on a new minibatch of the dataset. The other inputs are the target evaluation $\psi$, scalar width $\lambda$, learning rate schedule $\eta$, sample size $S$, and desired confidence $\beta$. In practice, we fixed $\lambda$ to be 1% of the training objective at the MLE, though it is also possible to average the IDM output across variable $\lambda$.

We use TensorFlow v2.1.0 `tensorflow.compat.v1` to compute the gradients of the models in all experiments [1]. In experiments with a Gaussian process (GP), we applied the GPFlow package and optimized the marginal likelihood w.r.t. the kernel parameters (type II maximum likelihood) [3].

**1D Synthetic Experiments**   The setup for the synthetic experiments is as follows,

- **quadratic**: draw 25 data points $x \sim \mathcal{N}(0, 1)$ and let $y = \frac{1}{10}x^2 - \frac{1}{2}x + 5 + \frac{1}{10}\epsilon$, where $\epsilon \sim \mathcal{N}(0, 1)$. Use constant learning rate $\eta(\cdot) = \frac{1}{100}$.

**Algorithm 2:** Stochastic gradient finite-difference implicit delta method

**Input:** Stochastic objective $\tilde{\mathcal{L}}$, evaluation $\psi$, scalar $\lambda$, schedule $\eta$, sample size $S$, confidence $\beta$

1 **Function** SG-FDIDM($\tilde{\mathcal{L}}$, $\psi$, $\lambda$, $\eta$, $S$, $\beta$):
    // randomly initialize parameters
2     $\hat{\theta}_n \sim \text{Uniform}(-0.1, 0.1)$
    // maximize $\mathcal{L}$ with stochastic gradient ascent
3     $i \leftarrow 0$
4     **while** $\hat{\theta}_n$ *not converged* **do**
5         $\hat{\theta}_n \leftarrow \hat{\theta}_n + \eta(i)\nabla_\theta \tilde{\mathcal{L}}_i$
6         $i \leftarrow i + 1$
7     **end**
    // collect $S$ samples for estimating $\psi(\hat{\theta}_n)$
8     **forall** $s \in [0, S)$ **do**
9         $\psi_{0,s} \leftarrow \psi(\hat{\theta}_n)$
10        $\hat{\theta}_n \leftarrow \hat{\theta}_n + \eta(i)\nabla_\theta \tilde{\mathcal{L}}_i$
11        $i \leftarrow i + 1$
12    **end**
    // maximize $\mathcal{L} + \lambda\psi$ with stochastic gradient ascent
13    **while** $\hat{\theta}_n$ *not converged* **do**
14       $\hat{\theta}_n \leftarrow \hat{\theta}_n + \eta(i)\nabla_\theta(\tilde{\mathcal{L}}_i + \lambda\psi)$
15       $i \leftarrow i + 1$
16    **end**
    // collect $S$ samples for estimating $\psi(\hat{\theta}_n(\lambda))$
17    **forall** $s \in [0, S)$ **do**
18       $\psi_{\lambda,s} \leftarrow \psi(\hat{\theta}_n)$
19       $\hat{\theta}_n \leftarrow \hat{\theta}_n + \eta(i)\nabla_\theta(\tilde{\mathcal{L}}_i + \lambda\psi)$
20       $i \leftarrow i + 1$
21    **end**
    // take average of samples
22    $\bar{\psi}_0 \leftarrow \frac{1}{S}\sum_{s \in [0,S)} \psi_{0,s}$
23    $\bar{\psi}_\lambda \leftarrow \frac{1}{S}\sum_{s \in [0,S)} \psi_{\lambda,s}$
24    **return** $\bar{\psi}_0 \pm \Phi^{-1}((1+\beta)/2)\sqrt{\frac{1}{\lambda}(\bar{\psi}_\lambda - \bar{\psi}_0)}$     // confidence interval for $\psi(\hat{\theta}_n)$

- **sin**: fix 160 data points evenly spaced in ranges $[-1.5, -0.7)$ and $[0.35, 1.15)$ and let $y = -\sin(3x - \frac{3}{10}) + \frac{1}{10}\epsilon$, where $\epsilon \sim \mathcal{N}(0, 1)$. Use constant learning rate $\eta(\cdot) = \frac{1}{200}$ for MLE objective and $\frac{1}{2000}$ for $\psi$-regularized objective.

In each case, the objective is evaluated over the entire dataset, such that $\tilde{\mathcal{L}} = \mathcal{L}$ and $S = 1$. In accordance with a univariate Gaussian density over the prediction, the network is trained to minimize half the sum of squared errors and the resulting interval is divided by the root mean squared error after convergence. The visualized covariance matrices were projecting to ensure positive semi-definiteness. This was done by averaging the matrix with its transpose, finding the eigendecomposition of the resulting matrix, then reconstructing the matrix with only the eigenvectors corresponding to positive eigenvalues.

**Benchmark Classification Experiments** For all classification datasets, the stochastic objective $\tilde{\mathcal{L}}$ is defined over minibatches of size $B = 128$. $\tilde{\mathcal{L}}_i$ is selected by stepping repeatedly through the randomly shuffled dataset in blocks of size $B$. RMSProp with an initial learning rate of $0.01$ is used for optimization.[7] $\psi$ is evaluated over a held-out validation dataset, a randomly selected 20% subset of the data for the benchmarks except mnist which is evaluated on the standard 10k test split of the 60k-10k partition.

---

[7] http://www.cs.toronto.edu/~tijmen/csc321/slides/lecture_slides_lec6.pdf

**Algorithm 3:** Multivariate finite-difference implicit delta method

**Input:** Learning objective $\mathcal{L}$, multivariate evaluation $\psi$, scalar $\lambda$

1 **Function** MV-FDIDM($\mathcal{L}$, $\psi$, $\lambda$):
2      $\hat{\theta}_n \leftarrow \arg\max_\theta \mathcal{L}(\theta)$            `// optimize learning objective`
3      $K \leftarrow \dim(\psi(\theta))$
4      **forall** $k = 1, \dots, K$ **do**
5          $\hat{\theta}_n^{(k)}(\lambda) \leftarrow \arg\max_\theta \mathcal{L}(\theta) + \lambda\psi_k(\theta)$      `// optimize` $\psi_k$`-regularized objective`
6      **end**
7      **return** $\left[ \frac{1}{\lambda}(\psi_j(\hat{\theta}_n^{(k)}(\lambda)) - \psi_j(\hat{\theta}_n)) \right]_{j=1:K, k=1:K}$      `//` $K \times K$ `covariance matrix of` $\hat{\psi}_n$

**KL Down-Weighting Experiment**    We use the standard 60k `mnist` train dataset. $\psi$ is defined as the total negative reconstruction loss on a random 1k subset of the training data. The Adam [2] optimizer is applied to the (possibly) regularized VAE objective with initial learning rate 0.001. The experiment adapted the VAE implementation in TensorFlow available at https://github.com/wiseodd/generative-models.

## D    Finite-Difference IDM for Multivariate $\psi$

In this section we present the multivariate algorithm for finite-difference IDM (FDIDM). In Alg. 3, a multivariate evaluation function $\psi$ is taken as input, along with a learning objective $\mathcal{L}$ and scalar width $\lambda$ (as in the univariate case). The algorithm optimizes the original MLE objective, then optimizes $K$ regularized MLE objectives, each with a different dimension of $\psi$. To obtain the estimated $K \times K$ covariance matrix $\hat{\Sigma}$ of $\psi$, finite differences are calculated by crossing the output dimension of $\psi$ corresponding to the row of $\hat{\Sigma}$ with the optimized $\psi$-regularized objective corresponding to the column of $\hat{\Sigma}$.

## Supplementary Material References

[1] Martín Abadi, Ashish Agarwal, Paul Barham, Eugene Brevdo, Zhifeng Chen, Craig Citro, Greg S. Corrado, Andy Davis, Jeffrey Dean, Matthieu Devin, Sanjay Ghemawat, Ian Goodfellow, Andrew Harp, Geoffrey Irving, Michael Isard, Yangqing Jia, Rafal Jozefowicz, Lukasz Kaiser, Manjunath Kudlur, Josh Levenberg, Dandelion Mané, Rajat Monga, Sherry Moore, Derek Murray, Chris Olah, Mike Schuster, Jonathon Shlens, Benoit Steiner, Ilya Sutskever, Kunal Talwar, Paul Tucker, Vincent Vanhoucke, Vijay Vasudevan, Fernanda Viégas, Oriol Vinyals, Pete Warden, Martin Wattenberg, Martin Wicke, Yuan Yu, and Xiaoqiang Zheng. TensorFlow: Large-scale machine learning on heterogeneous systems, 2015. URL https://www.tensorflow.org/. Software available from tensorflow.org.

[2] Diederik P Kingma and Jimmy Ba. Adam: A method for stochastic optimization. *arXiv preprint arXiv:1412.6980*, 2014.

[3] Alexander G. de G. Matthews, Mark van der Wilk, Tom Nickson, Keisuke. Fujii, Alexis Boukouvalas, Pablo León-Villagrá, Zoubin Ghahramani, and James Hensman. GPflow: A Gaussian process library using TensorFlow. *Journal of Machine Learning Research*, 18(40): 1–6, apr 2017. URL http://jmlr.org/papers/v18/16-537.html.

[4] Whitney K Newey and Daniel McFadden. Large sample estimation and hypothesis testing. *Handbook of econometrics*, 4:2111–2245, 1994.

[5] Aad van der Vaart and Jon Wellner. Weak convergence and empirical processes. *Springer*, 1996.