# OpenReview forum: "The Implicit Delta Method"
_NeurIPS.cc/2022/Conference — NeurIPS 2022 Accept_

### Official Review · Reviewer_41Kr · 2022-06-30

**Rating:** 7
**Confidence:** 4
**Soundness:** 2 fair
**Presentation:** 4 excellent
**Contribution:** 4 excellent

**Summary:**

This paper focuses on estimating sampling uncertainty for typical machine learning models (M-estimators) with some evaluation function $\psi$ (e.g. the test loss). They do so by using the delta method to estimate the asymptotic variance of $\psi$. While this is not a new idea, the authors note that there are two downsides to the traditional use of the delta method: (1) if the model has parameters of dimension $D$, it requires the inversion of a $D \times D$ matrix, and (2) if $\psi$ is non-differentiable, then the delta method does not apply. The authors show that solving a regularized version of the M-estimator can get around both (1) and (2).

**Questions:**

- The discussion in Remark 1 seems to imply that the loss needs to be the log likelihood of some probability distribution (otherwise I'm not following why we care about $\sigma$). Why is this? It seems like a lot of the arguments in the paper go through as long as $\hat\theta_n$ is any M-estimator.

- In the statement of Theorem 4, $m$ has to satisfy $m \asymp n$ or $m \gg n$. Could this be more concisely stated as $m$ is $\Omega(n)$?

- Is Theorem 1 not just proving conditions under which $\hat V_n^{IIDM} = \hat V_n^{DeltaMethod}$? If true, this seems like a more precise statement of the theorem.

- This is a much broader question. How do the authors think such local approximations should be used when the model objective is multimodal? Should we just be approximating around the discovered mode? Should we discover multiple local optima and then run this kind of approximation for each optimum? I don't think it's necessarily the current authors' job to solve such questions, but it could be worth commenting on in the paper if they have insights.



**Limitations:**

Yes, I think the authors have addressed the potential negative impact of their work.

**Strengths And Weaknesses:**

**Strengths**
1. Issues (1) and (2) I mentioned in the summary are really big problems with previous work in estimating uncertainty in large / complex machine learning models, and I think the authors make a pretty convincing case that they have solved these issues. I actually think the authors are underselling the "you don't have to invert a big Hessian matrix" thing. A number of people have tried to deal with this issue for related approximations (i.e. infinitesimal approximations involving the inverse Hessian) and been unsuccessful. Koh and Liang 2017 use a stochastic approximation to the inverse Hessian. But Stephenson et al. 2020 find it poor for their more quantitative purposes (I suspect the current authors would find it unusable for their implicit delta method as well) and propose a different approximation. But Frangella et al. 2021 then show that the approximation of Stephenson et al. 2020 is flawed and propose yet another approximation. Their analysis might be the final word in this chain, but who knows! So, it's really great to not have to deal with solving systems with large Hessians.
2. The writing quality in the paper is great (significantly above average of the typical NeurIPS submission).

**Weaknesses**

I think the current empirical evaluation needs to be significantly strengthened. I have a few points:
1. To me, the minimal thing the experimental section should show is whether or not the proposed method improves over baselines in some way. If someone asked me to point them to parts of this paper for concrete evidence why they shouldn't just always run the bootstrap or the standard delta method to estimate sampling uncertainty, I wouldn't know what parts to point them to. I would be surprised if the authors couldn't demonstrate that their IDM is faster than the bootstrap and, in high dimensions, the delta method. Or show a model on which IDM runs but the delta method does not due to lack of differentiability. But this needs to be actually shown (or, if this isn't true, then it should be investigated why a good-sounding idea doesn't work in practice).
2. I'm not sure what to take away from Fig 3; I can't tell from this figure whether IDM is doing something good or bad. First, why should we trust the bootstrap here (how do we know both the bootstrap and IDM aren't just terrible here -- could one argue that the dimension of the model is too high compared to the amount of training data for any asymptotic estimate of variance to have kicked in)? Second, assuming we do trust the bootstrap, why does this figure say that we should trust IDM? There are differences between IDM and the bootstrap. In what sense are these large or small? It's a little hard to tell, since $\psi$ doesn't relate to any particular downstream task. It might be more helpful if $\psi$ were something more meaningful. Also, how fast is IDM versus the bootstrap here? If IDM takes 500 years to run, I sure would rather run the bootstrap!
3. On the synthetic example: again, it's hard to tell if IDM is actually doing something sensible here -- is it closer (or reasonably close compared to other methods) to the true sampling uncertainty? Since you have access to ground truth here, this could be a great opportunity to give a much more quantitative assessment.
4. Why are Gaussian processes included in the experiments? Bayesian methods don't measure sampling uncertainty, so what is being gained by comparing them to a method for approximating sampling uncertainty?

Overall, I think this could be a really great paper if the evaluation section were stronger. I would definitely raise my score and push for an accept if the authors can provide more compelling experiments that carefully investigate whether or not there are practically meaningful regimes in which their method is an improvement (e.g. is much faster without much loss in accuracy).

**Smaller comments**

- On line 242, [13] is cited as an example of approximating the bootstrap with similar infinitestimal approximations. I wouldn't cite this paper as an example of this, as it doesn't really work for the bootstrap. You might cite Giordano et al. 2019 instead (see their  Proposition 4 about the bootstrap).
- Line 225: I don't think you only need a small number of stochastic gradient steps; this should be true for pretty much any optimizer.
- Eq (5): I would think of the version of Eq (5) using the empirical estimate of the delta method estimator as more standard in practice. This also fits better with the following sentence which says Eq (5) won't work well if $\theta$ has many dimensions (if we're actually using the population version, then it won't work well in way more cases because $I$ is hard to compute!)
- Line 135 "it is still not clear how to do [the differentiation needed to compute the delta method] in practice". I think this is clear -- we can use the implicit function theorem, which is exactly what the proof of Theorem 1 is doing.
- I think the experiments in Fig 2 should be changed (see comments above), but at the very least the scale of the color bars should be the same so we can compare the different methods.


Z. Frangella, J. A. Tropp, and M. Udell. Randomized Nystrom Preconditioning. arXiv. 2021.

R. Giordano, M. I. Jordan and T. Broderick. A Higher-Order Swiss Army Infinitesimal Jackknife. arXiv. 2019.

P. W. Koh and P. Liang. Understanding Black-box Predictions via Influence Functions. ICML. 2017.

W. T. Stephenson, M. Udell, and T. Broderick. Approximate cross-validation with low-rank data in high dimensions. NeurIPS. 2020.

---

> ### Author Response · Authors · 2022-08-02
> **Response to Reviewer 41Kr**
>
> ## Strengths
> 1. Thank you for recognizing the strength of our contribution, and thank you for pointing us to some great references to better highlight and contextualize the value of our contribution. We will add them! Thank you.
>
> 2. Thank you; we worked hard to communicate the technical material clearly and we are glad it showed.
>
> ## Weaknesses
> 1\. This is a great point! Thank you for the suggestion on how to more strongly highlight our contribution. We will add both theoretical and experimental details on computational cost; see the table below. In sum, the additional detail will more clearly support the motivation for IDM.
>
> Experimentally, we have recorded the runtimes of IDM and bootstrap for the experiment in Fig 3. The following new table reports the runtime in seconds:
>
> |Runtime (sec)|Vehicle|Waveform|Satellite|MNIST|
> |---|---|---|---|---|
> IDM|39|129|111|303|
> Bootstrap|806|2334|3192|7164|
>
> We will repeat this a few more times and report average runtimes with confidence intervals as well as detail the machine this ran on. This can serve as that table one can directly point to, that you mentioned. We will also note that we only used 50 resamplings for bootstrap, which is honestly on the low end -- we did so only to be able to run the experiment in a reasonable amount of time -- all bootstrap runtimes double if we use 100, increase 20-fold if we use 1k. Also, we can potentially make IDM even more efficient by warm starting the regularized optimization after solving the unregularized MLE -- we didn't bother only because it was already fast enough to run easily, but this could reduce runtimes by at most one half.
>
> Theoretically, FDIDM has time complexity $\mathcal{O}(C K)$,
>  where $C$ is the cost to perform MLE a single time (often the dominating cost) and $K$ is the dimension of the image of $\psi$ (usually $K=1$). Compared to delta method, an important feature here is that dimensionality of the parameters $\theta$ is completely implicit. In short, as long as it is tractable to optimize the original model, it is tractable to estimate the covariance matrix of $\hat{\psi}_n$. Theoretically, there is of course a relationship between the hardness of optimization and dimensionality, but the point is that here it is completely implicit, rather than explicitly having to compute and invert large second-order derivative matrices. This is important because we often use first-order optimization to solve the MLE. We will make this point clearer. Bootstrap has time complexity $\mathcal{O}(C B)$ where $B$ is the number of resamplings we draw, which generally is chosen around 50--100 $\times K$ (we need to scale with $K$ to avoid a rank-deficient estimated covariance).
>
> 2\. Indeed, since the ground truth is unknown, Fig. 3 is less a benchmark of performance and more a demonstration of usage and that IDM is giving reasonable numbers not too far off from what the bootstrap would give (even with just 50 resamplings), with far faster computation (as the above runtimes show). We will make this characterization of the experiment explicit. Regarding interpretation of differences as reasonable, as a cross entropy, $\psi$ does have units (bits), and one could compare the scale of differences to the scale of the point predictions.
>
> 3\. We agree. As suggested in the response to PMCG, we will add a plot showing the mean-squared error of estimated variance vs true sampling variance for each method. (In addition to for FDIDM with different choices of $\lambda_n$.)
>
> 4\. We agree. We included it only for reference as GPs are often used to understand uncertainty in ML. But indeed theoretically the model and guarantees are fundamentally different (not frequentist), and this may cause confusion. We suggest to move the GP plots to the appendix, where we can also provide additional context.
>
> ["Smaller comments" and "Questions" answered in next reply due to character limit]

---

> > ### Comment · Reviewer_41Kr · 2022-08-06
> > **Reply**
> >
> > Thanks for the very detailed responses to all of the reviews; I think what you've said here makes a lot of sense, and that the comparisons of runtimes in Fig 3 along with quantitative reporting of MSE in a new Fig 2 (rather than its current qualitative form) address my main issues about the paper as-is. I've increased my score from a 6 to a 7.
> >
> > Reading the responses, discussion, and other reviews, I had a couple extra suggestions:
> > 1. It would be good to clearly discuss what kind of uncertainty the paper is about right at the beginning and keep coming back to this throughout the paper. There are so many different kinds of "uncertainty" that are discussed at NeurIPS and related venues (e.g. uncertainty from the model being wrong, uncertainty from some testing data being out-of-distribution, uncertainty from not having enough data), and I think it's often easy to get them mixed up. E.g. as JWsG has noted, the current abstract doesn't exactly scream "this paper is about sampling uncertainty," especially since it starts out talking about predictions. I'd also note that "[getting credible intervals for predictions] is solved by the delta method" isn't necessarily true -- this is true if the model is right and the points your predicting on are in distribution. An editing pass to really emphasize (1) this paper is about sampling uncertainty and (2) what sampling uncertainty does and does not measure, would be really helpful.
> > 2. I think the suggested changes to the experiments are sufficient for acceptance. But I think even more could be better. It would be great to demonstrate: (1) that the mean squared error to the true sampling uncertainty (estimated using repeated draws in a synthetic experiment) decreases with n, (2) the computation time of the proposed method also beats the computation time of the exact delta method in situations where the approximation error is also low (e.g. like a $50,000$-dimensional experiment where the implicit delta and exact delta methods are shown to compute basically the same thing, but it takes a really long time to invert the Hessian matrix).
> >
> > In any case, looking forward to reading the updated paper!

---

> > > ### Author Response · Authors · 2022-08-09
> > > **Response to Reviewer Response**
> > >
> > > Thank you for the encouraging words. We are glad you found the response helpful.
> > >
> > > Thank you for the helpful pointers regarding clarifying the terminology on uncertainty in the paper and demonstrating the value of IDM experimentally which we will follow. As mentioned in our latest reply to Reviewer PMCG, we have now produced an additional plot showing the mean squared error to true sampling uncertainty as you proposed. Please see Figure 7 in Appendix C in the revised submission and surrounding text. We tried various values of $\lambda$ and found that the performance is very insensitive to this choice. This is what we were able to put together this week and we will certainly refine these for inclusion in a final version.

---

> ### Author Response · Authors · 2022-08-02
> **Response to Reviewer 41Kr, continued**
>
> (continued ...)
>
> ## Smaller comments
>
> - Agreed. Thank you!
> - Agreed. We will revise.
> - We agree this is subjective. Most introductory texts on MLE inference use the form in Eq. (5) (see e.g. Wasserman [32]), which is why we chose that presentation, while for conditional MLE it is standard to at least average over the empirical rather than theoretical $X$ distribution. We will add a comment that another possible estimator for $I(\theta_0)$ other than $I(\hat\theta_n)$ is $-\frac1n\sum_{i=1}^n\nabla^2\log f(Z_i;\hat\theta_n)$.
> - Fair. We will revise.
> - We agree. We'll defer the heatmaps to the appendix and instead use the space to show MSE plots for sampling-variance estimation, with a line for each method, on the same plot.
>
> ## Questions
>
> - The extension to general $M$-estimation is actually non-trivial as we implicity rely on the Information Matrix Equality, which only applies in MLE. We will mention this in the conclusions.
> - Sure. We will revise.
> - No; not in the form of Eq. (5). Only in the form of the empirical Fisher information estimator. We'll define it next to Eq. (5) and refer to it in Thm 1.
> - This indeed somewhat beyond our scope, as it violates the conditions of delta method, but our non-rigorous intuition says that as long as the distribution of $\psi(\theta)$ looks the same for $\theta$ around each trough in the loss surface then the estimate of the variance of $\hat\psi_n$ that we obtain from just looking at one trough will be the same if we (more correctly) looked locally near each trough. This is relevant to NNs when $\psi$ only depends on $\theta$ via model evaluations because often multimodality comes about due to the predictions being the same for different weights.

---

### Official Review · Reviewer_JWsG · 2022-07-11

**Rating:** 5
**Confidence:** 3
**Soundness:** 3 good
**Presentation:** 3 good
**Contribution:** 3 good

**Summary:**

Authors propose a method for uncertainty estimation for predictive models. This method aims to directly compute uncertainty estimates for the cost function used to evaluate the predictive model. Main advantage of this method is that it avoids uncertainty quantification of the model parameters. This work provides consistency results in the large sample limit and it is empirically evaluated on different predictive tasks.


**Questions:**

Could you please provide empirical evidence comparing your approach to a (non-bootstrapped) ensemble of neural networks?

Why don’t you also employ the Fashion-MNIST data set and a simple neural network like Lenet-5? If you want to focus on UCI-data sets, the set of selected data sets is very reduced.

Which are the benefits of your approach against other approaches such as conformal predictions? This discussion should be added to the paper.



**Limitations:**

I ask authors to discuss the problem of not capturing the multimodality of the log-likelihood function.


**Strengths And Weaknesses:**

Strengths:

- The approach provides an alternative approach for uncertainty quantification which circumvents the problem of computing uncertainty estimates over the model parameters. This could potentially be a very interesting research direction, because modelling parameter uncertainty has proven to be extremely challenging in many models, especially neural networks.

 - The approach is theoretically motivated by some consistency guarantees in the large sample limit.

- Usually, uncertainty quantification is  computationally very expensive. The proposed method provides a much cheaper way to compute confidence intervals.

- Although the approach builds on previously widely known results, it’s adaptation to more general and larger models is novel and interesting.


Weaknesses:

- The only uncertainty that this approach is able to capture is the uncertainty around the exact location of the MLE estimate. In the case of a model with a highly multimodal loss landscape, the proposed method will be able to compute the uncertainty which belongs to a single mode. In models like NNs, it is widely known that capturing this kind of “local” uncertainty is much less powerful than capturing uncertainty involving multiple modes of the loss landscape, as it is done by ensemble of NNs or by powerful stochastic-gradient-Monte-Carlo methods:

B. Lakshminarayanan, A. Pritzel, and C. Blundell, “Simple and scalable predictive uncertainty estimation using deep ensembles,” in Advances in neural information processing systems, 2017, pp. 6402–6413.

Wenzel, Florian, et al. "How good is the Bayes posterior in deep neural networks really?." arXiv preprint arXiv:2002.02405 (2020


*** Updated after discussion with authors ***
I have updated this part of the review after the discussion with other authors and reviewers.

---

> ### Author Response · Authors · 2022-08-02
> **Response to Reviewer JWsG**
>
> ## Strengths:
>
> Thank you for recognizing both the importance of the problem we study and the strength of our contribution.
>
> ## Weaknesses:
> (answered in order)
>
> - While correct that our guarantees are asymptotic, we believe this comment misunderstands the purpose of inference based on asymptotic normality, as we do here and as is done generally in MLE and delta method (and also all the NeurIPS papers mentioned in the response to Reviewer ZZii).
>
>   - First, applying in the large sample limit is a _feature, not a bug_. Only in the large sample limit can one hope to characterize distributions _exactly_ and therefore get calibrated confidence intervals with coverage _exactly_ 95%. Finite sample guarantees are usually too conservative by necessity. Consider for example a confidence interval on the mean of a Bernoulli random variable with success probability $p\in(0,1)$ constructed by adding some $\pm \Delta$ around the sample mean $\hat p_n$ from $n$ iid draws. The standard MLE confidence interval is $\hat p_n\pm 1.96\sqrt{\hat p_n(1-\hat p_n)/n}$, which has coverage approaching _exactly_ 95%. If instead we invert the exact finite-sample guarantee from Hoeffding's inequality, we get $\hat p_n\pm1.36/\sqrt{n}$, which is guaranteed to have coverage at least 95\% for each $n$ by Hoeffding, but has coverage converging to $\mathrm{erf}(0.96/\sqrt{p(1-p)})$ as $n\to\infty$, which is 0.993 for $p=1/2$ and 0.99996 for $p=1/8$, i.e., it is far far too big in practice. This calibration is why confidence intervals are usually constructed using asymptotic normal approximations rather than finite-sample guarantees.
>
>   - Second, inference on (functions of) model parameters and conformal prediction answer two fundamentally different questions. The former provides uncertainty quantification on the model being learned while the later provides prediction bands for the observed outcomes. Taking a very simple example for illustration purposes only, if we consider a classic regression model $Y=g(X;\theta)+\epsilon$, MLE provides inference on $\theta$ (and functions thereof via the delta method) and conformal prediction provides intervals on the outcomes $Y_i$ themselves. In particular, while the former will give intervals that converge to a point (focusing only on the vanishing epistemic uncertainty), the latter will always remain an interval due to the fundamentally unpredictable (aka aleatoric) uncertainty in $\epsilon$. Thus, these are just fundamentally two different beasts with different purposes. Of course, an important appeal of conformal prediction is that it does not actually depend on the model and how we predict $Y_i$, as it treats this as a blackbox and simply does inference on $Y_i-\hat Y_i$. But if we want to make inferences on the model itself, we cannot treat it as a blackbox, by definition. We will certainly add some discussion of conformal prediction for context, but, in sum, it's just a very different problem.
>
> - You are correct that the implicit delta method (IDM) assumes the same regularity conditions as the delta method (DM), and therefore has the same theoretical limitations (except for a small extension given in Sec 3.2, where $V_n^{\mathrm{DeltaMethod}}$ is ill-defined). This a basic feature of the setting, and as explained in the response to Reviewer PMCG, while we do specify this, we can make this extra clear expositionally. The focus of the paper is on reducing the significant computational burden of DM in large models. There are reasons to believe DM (and hence) IDM should continue to work in multimodal settings as long as $\psi$ is largely invariant to this multimodality (e.g., depends on $\theta$ only via model evaluation and multimodality arises from equal predictions with different parameters); we can comment on this, but this is outside the scope of our rigorous theoretical results; see also response to last question by Reviewer 41Kr. It is also worth noting that the methods you mention are potentially computationally burdensome and are also specific to NNs.
>
> [continued in next reply due to character limit]

---

> > ### Comment · Reviewer_JWsG · 2022-08-05
> > **Clarifications of my previous review**
> >
> > Dear Authors,
> >
> > Thanks for your detailed response. I try to clarify some parts of my review.
> >
> > 1. I partly agree with the misunderstanding about the problem you are trying  solve and some of my criticisms of this work. The first two lines of the abstract state: "Uncertainty quantification is a crucial part of drawing credible conclusions from predictive models, whether concerned about the prediction at a given point or any downstream evaluation that uses the model as input". The problem I mainly refer during my review was about uncertainty quantification about "the prediction at a given point". This same problem is mentioned by authors at Line 41 and Lines 76-77. Section 5.1. is devoted to this problem too. I was confused with the experimental settings of Section 5.2, because I got the impression it was also about uncertainty quantification about "prediction at a given point". I ask you to better clarify this in the experimental settings because they are very similar to the experimental settings of many works about uncertainty quantification in neural network predictors where, by default, the main focus is to quantify uncertainty about "the prediction at a given point".
> >
> > 2. My comment about theoretical guarantees being valid only in the large sample limit is that, even though you provide a confidence interval, there is no way in real-life settings to guarantee the correct coverage of the confidence interval computed by your method.  IMO, this is a weakness of your method. For predictions at given point, conformal methods can provide such guarantees. But this limitation is something which is not discussed in Section 6.
> >
> > 3. All your evaluation focuses on NNs. NNs are known to have a highly-multimodal likelihood function. When quantifying uncertainty about "the prediction at a given point", there is large evidence in the literature that capturing this multimodality is key in NNs. You mention this issue in your literature review. The question is that your method only captures parameter uncertainty at a local level. In consequence,  we can expect that the confidence intervals computed by your method will be affected by this issue, even in the case of confidence intervals for general cost-functions in downstream tasks. I agree that your approach capture some uncertainty, but, IMO, according to the existing literature, a lot of uncertainty will be lost. And this is something which is not properly discussed in this work.

---

> > > ### Author Response · Authors · 2022-08-06
> > > **Re-response to Reviewer JWsG clarifications**
> > >
> > > Thank you for reading and engaging with our reply, and thank you for agreeing, albeit partly, with our response. Let us respond to your clarifications so we can hopefully arrive at a consensus. On the whole, we agree with you that some exposition/discussion can be improved, and we thank you for pointing this out and helping us improve the paper, but we do not think the value of our contribution is affected.
> > >
> > > 1. We will clarify earlier on what we mean by uncertainty quantification at a given point in the experimental settings, thank you for the suggestion. In more detail, by “uncertainty quantification at a given point” for a prediction model $g_\theta(x)$ parameterized by $\theta$, we mean uncertainty quantification for the evaluation function $\psi(\theta)=g_\theta(x_0)$ where $x_0$ is a fixed (i.e., given) input, as specified on line 76 (but can certainly be clarified earlier and more clearly). This is just a function of the model. Good confidence intervals for this quantity (a fixed but unknown scalar) must shrink to a point. In contrast, good predictive intervals *for the response* (such as given by conformal prediction) should not shrink to a point, as they must account for aleatoric uncertainty in the unpredictable noise.
> > >
> > > 2. We agree that the limitations of asymptotically calibrated frequentist confidence intervals (which are **not** unique to our method) are worth mentioning in Section 6. To reiterate, we also agree that adding a comparison to conformal prediction (which solves a different problem) would strengthen the paper, and we will add it to Section 4. We, nonetheless, again emphasize that asymptotic-ness is a basic feature of this type of statistical inference, and the criticism you voice equally applies to the standard 95%-confidence interval for a mean made by adding/subtracting $1.96/\sqrt n$ sample standard deviations from the sample mean.
> > >
> > > 3. We agree that more discussion around the context of the work is merited, but that this is limited to expositional modifications. Essentially, we have some theory that is rigorous but is rooted in high-level regularity conditions (the same as the delta method), and motivated by this theory we apply our method generally without necessarily checking these conditions. While this is very common in statistical methodology and often done implicitly, that is certainly no excuse and we will make this connection between theory and application more explicit in our exposition. We will also discuss motivation for either why the conditions should approximately hold or to what extent we think that our and related methods should still work when the conditions fail (see for example our answer to Reviewer 41Kr's last question).

---

> ### Author Response · Authors · 2022-08-02
> **Response to Reviewer JWsG, continued**
>
> ## Weaknesses: (continued)
> (here answering the last two bullets under Weaknesses)
>
> - Again, we think there is some misunderstanding of the nature of the problem we study. Nonetheless, this feedback is useful for us to better clarify what we are doing. The "bootstrap" we refer to is a frequentist inference method to estimate sampling variance and construct confidence intervals, see https://en.wikipedia.org/wiki/Bootstrapping_(statistics)#Methods_for_bootstrap_confidence_intervals and Ch. 8 of Wasserman [32]. In particular, the prediction is still done with a neural net trained on the whole data. We only wish to estimate the distribution of this prediction over sampling of new random datasets. The bootstrap approximates it using resampling. In contrast, it appears that the "ensemble of boostrapped neural networks" you refer to is bootstrap aggregation (aka bagging), where the actual prediction itself is made by averaging many neural nets. That is what is used in Nixon et al. It is therefore not correct to say that "authors compared an ensemble of boostrapped neural networks." We will use this feedback to make far clearer what we mean by "bootstrap" and provide specific appropriate references regarding bootstrap confidence intervals, along with citations to the papers you mention to draw the contrast clearly.
>
> ## Questions
>
> - "Could you please provide empirical evidence comparing your approach to a (non-bootstrapped) ensemble of neural networks?"
>
> As mentioned above, we are not using bootstrap ensembles for prediction, just vanilla neural nets. The "bootstrap" here is only used for confidence interval construction.
>
> - "Why don’t you also employ the Fashion-MNIST data set"
>
> Thank you for the suggestion. Of course, we could only have included certain datasets and architectures, and there will always be some omission to be pointed out or a preference for another dataset. But we nonetheless agree that yours is a really great suggestion and we will add Fashion-MNIST with Lenet-5. This is indeed straightforward to do with the (provided) replication code.
>
> - "Which are the benefits of your approach against other approaches such as conformal predictions?"
>
> As discussed above, conformal prediction solves a different problem altogether. We will add a discussion comparing the two fundamentally different types of inference. The first-order bit is that we (and other model inferences like MLE, debiased lasso, etc.) focus solely on epistemic uncertainty and on the model being learned and _how_ it is learned, while conformal prediction incorporates both epistemic and aleatoric uncertainty and focuses on the responses to predicted and the blackbox prediction outputs.

---

### Official Review · Reviewer_PMCG · 2022-07-11

**Rating:** 6
**Confidence:** 4
**Soundness:** 3 good
**Presentation:** 3 good
**Contribution:** 3 good

**Summary:**

The paper proposes the _implicit delta method_ for estimating the asymptotic variance of an estimator of the form $\hat\psi_n = \psi(\hat\theta_n)$, where $\hat\theta_n$ is the MLE and $\psi: \Theta \to \mathbb{R}$ is a functional. The implicit delta method is computationally efficient in situations where fitting a model is computationally burdensome and/or the target functional is not known explicitly. The proposed method works by solving an additional estimation problem regularized with $\lambda \psi(\theta)$. The paper shows that subject to regularity conditions, $\hat V^{\mathrm{FDIDM}} = \lambda^{-1} (\psi(\hat\theta_n(\lambda)) - \hat\psi_n)$, where $\hat\theta_n(\lambda)$ is the regularized estimate resulting from the additional estimation problem, consistently estimates the asymptotic variance of $\hat\theta_n$.

**Questions:**

1. For the finite-difference IDM, one can be so free in the choice of $\lambda$: Isn't there a benefit in making $\lambda$ as small as possible without causing numerical instability? Judging by the proof of Theorem 2, it looks like the choice of $\lambda$ should depend on the curvature of $\hat\psi_n$ with respect to $\lambda$. Would it be possible to see experimental results using a grid of values of $\lambda$ for different problems?

2. Section 3.2 was more difficult to follow compared to other parts. I think it may help to start the section with concrete examples rather than with a general description. Also, it is said $\psi_0$ is random, but what is $\psi_0$ in this case? $\psi_0 = m^{-1} \sum_{j=1}^{m} h(W_j; \theta_0)$?

3. Figure 2 is not ideal for comparing the performances of different methods; the reader has to imagine putting one panel on top of another. Could you create a different visualization that highlights the differences more?

4. How do the methods compare in terms of computational cost?

**Limitations:**

The authors discuss some of the limitations of their work as a method but not the societal impact. Since this is a work in statistical methodology, I do not believe this is a defect.

**Strengths And Weaknesses:**

The paper proposes an intriguing alternative approach to computing the asymptotic variance that does not require inverting the Fisher information matrix (or an estimate thereof). The approach is shown to lead to a consistent estimate of the asymptotic variance, albeit under some regularity conditions.

One weakness of the paper is that the applicability of the method appears to be very much limited to the settings for which the delta method also applies. Thus, although the proposed approach works as a computationally lighter alternative to the delta method, it cannot be used with estimators that fall outside of the framework (e.g., lasso-type estimators or high-dimensional regimes). Perhaps it was not the intention of the authors to imply that their method can be applied to any estimation procedures, but this was the impression formed by reading earlier parts of the paper.

---

> ### Author Response · Authors · 2022-08-02
> **Response to Reviewer PMCG**
>
> ## Strengths And Weaknesses:
>
> - "One weakness of the paper is that the applicability of the method appears to be very much limited to the settings for which the delta method also applies"
>
> You are correct that the implicit delta method (IDM)
> assumes the same regularity conditions as the delta method (DM),
> and this is clearly stated in our theory (but for a small extension given in Sec 3.2, where $V_n^{\mathrm{DeltaMethod}}$ is ill-defined). We attempted to make this clear, for example, on line 89 and in Equation (2) (which, e.g., is not satisfied by lasso-type estimators) and by focusing on the Fisher information. We can absolutely make this clearer and explicit and more directly discuss the limitations of DM to begin with.
> In fairness, DM applies to a broad class of estimators,
> which we highlighted in the paper with several examples.
> In doing so, however, we did not mean to imply that IDM can be applied to any estimation procedure at all, and will make sure to clarify any potential for misunderstanding on this front.
>
> ## Questions:
> (answered in order)
>
> 1\. You are correct: Theorem 2 shows that finite-difference IDM (FDIDM) converges correctly for any choice of $\lambda_n$
> that's $o(n)$. The proof of Theorem 2 indeed proceeds by showing that $n\hat V_n^{\mathrm{FDIDM}}$ approximates $n\hat V_n^{\mathrm{IIDM}}$ up to an $O_p(\lambda_n/n)$ remainder term, which can of course be made smaller if we choose $\lambda_n$ smaller (see last display equation in the proof). And, we know that $n\hat V_n^{\mathrm{IIDM}}\to_p V_0$ by Theorem 1. Smaller $\lambda_n$ will lead to a better approximation of IIDM, but this _does not_ necessarily mean that smaller $\lambda_n$ is better, since the real target is $V_0$, not $n\hat V_n^{\mathrm{IIDM}}$, and not even $n\hat V_n^{\mathrm{DeltaMethod}}$. For example, when $\psi$ is only _approximately_ differentiable, finite differencing is _necessary_ (Sec 3.2). Moreover, there may be concerns regarding numerical instability, as you mention, just as numerically computing a derivatives using too small a finite difference may be unstable (e.g., below machine precision). So, in terms of revising exposition, we will (1) explain and discuss the above nature of Theorem 2 regarding the IIDM-approximating role of $\lambda_n$ and (2) provide references on numerical differentiation such as Ch. 4 of Numerical Analysis by Burden and Faires, as well as discuss the possibility of central differences and other formulae other than the forward differencing we used.
>
> In terms of experiments, we found that setting $\lambda_n$ to 0.01 worked well in practice and that performance was largely insensitive to this choice, but we agree we can make this message explicit and stronger. Since we actually compute the true sampling variance in our experiment in Figure 2, it is indeed a great setting for exploring explicitly the impact of the choice of $\lambda_n$ for estimating the true sampling variance (rather than for approximating the DM or IIDM variance estimators). We propose to add a plot showing the mean-squared error between predicted variance $\hat V_n^{\mathrm{FDIDM}}$ and true sampling variance as $n$ grows with a few lines, each for a different choice of $\lambda$ (either in the supplement or main text, depending on space constraints). This is indeed straightforward with the (provided) replication code.
>
> 2\. First, to answer your question, yes, $\psi_0=\psi(\theta_0)= m^{-1} \sum_{j=1}^{m} h(W_j; \theta_0)$, and it is random due to the randomness of $W_1,\dots,W_m$. We propose to explicitly remind the reader of this definition and highlight the randomness on line 185.
>
> Second, we will implement your suggestion and move the concrete example we currently give on lines 189--198 to the beginning of the section. We agree this will help with comprehension. We will also expand the non-specific ``Other non-differentiable examples'' on line 192 to specifically mention hinge loss for classification and check loss for quantile regression.
>
> 3\. Thank you for the suggestion. We will also add a similar mean-squared error plot as suggested in our answer to your first question with a line per method. Again, this is straightforward with the (provided) replication code.
>
> 4\. [see next reply due to character limit]

---

> ### Author Response · Authors · 2022-08-02
> **Response to Reviewer PMCG, part 2**
>
> ## Questions: (continued)
>
> 4\. Thanks for the great question. We will add both theoretical and experimental details on computational cost; see below. In sum, the additional detail will more clearly support the motivation for IDM.
>
> Theoretically, FDIDM has time complexity $\mathcal{O}(C K)$,
>  where $C$ is the cost to perform MLE a single time (often the dominating cost) and $K$ is the dimension of the image of $\psi$ (usually $K=1$).
> Compared to delta method, an important feature here is that dimensionality of the parameters $\theta$ is completely implicit.
> In short, as long as it is tractable to optimize the original model, it is tractable to estimate the covariance matrix of $\hat{\psi}_n$. Theoretically, there is of course a relationship between the hardness of optimization and dimensionality, but the point is that it is here completely implicit, rather than explicitly having to compute and invert large second-order derivative matrices. This is important because we often use first-order optimization to solve the MLE. We will make this point clearer. Bootstrap has time complexity $\mathcal{O}(C B)$ where $B$ is the number of resamplings we draw, which generally is chosen around 50--100 $\times K$ (we need to scale with $K$ to avoid a rank-deficient estimated covariance).
>
> Experimentally, we have recorded the run times of IDM and bootstrap for the experiment in Fig 3. The following new table reports the runtime in seconds:
>
> |Runtime (sec)|Vehicle|Waveform|Satellite|MNIST|
> |---|---|---|---|---|
> IDM|39|129|111|303|
> Bootstrap|806|2334|3192|7164|
>
> We will repeat this a few more times and report average runtimes with confidence intervals as well as detail the machine this ran on.
> We will note that we only used 50 resamplings for bootstrap, which is honestly on the low end -- we did so only to be able to run the experiment in a reasonable amount of time -- all bootstrap runtimes double if we use 100, increase 20-fold if we use 1k. Also, we can potentially make IDM even more efficient by warm starting the regularized optimization after solving the unregularized MLE -- we didn't bother only because it was already fast enough to run easily, but this could reduce runtimes by at most one half.

---

> ### Comment · Reviewer_PMCG · 2022-08-08
> **Response to Author Rebuttal**
>
> Thank you for answering my questions! After reading through all the reviews and the responses, I am of the opinion that the paper adds an interesting new tool for variance estimation and am inclined to argue for an accept. I strongly believe that the current draft is missing some important details (e.g., comparisons of computational costs, sensitivity with respect to the choice of $\lambda$) and the presentation can be improved in some places (e.g., the introductory section could clarify the applicable setup; Section 3.2 could be rewritten so that it is both easier to understand and better at highlighting an advantage that is unique to the method; more work can be done to make the presentation of the experimental results more appealing). However, these appear to be rather minor improvements that could not be difficult to implement.

---

> > ### Author Response · Authors · 2022-08-09
> > **Response to Reviewer Response**
> >
> > Thank you for reading and engaging with our response. And thank you for your support for acceptance. We agree that some improvements are needed and that these are minor and do-able within the scope of preparing a camera ready if the paper is accepted. To increase confidence that this is doable, that we will do it, and that the results are favorable, we shared initial runtime experiments and theoretical characterization in our response, and we can now also share initial experiments regarding the choice of $\lambda$ (see the new Figure 7 in updated Appendix C and surrounding text). These is just what we were able to put together this week and we will further refine these for inclusion in a final version. The experiments on varying $\lambda$ confirm that one should use a moderate value of lambda and that, in the end, the performance is very insensitive to this choice.

---

### Official Review · Reviewer_ZZii · 2022-07-15

**Rating:** 8
**Confidence:** 1
**Soundness:** 3 good
**Presentation:** 3 good
**Contribution:** 4 excellent

**Summary:**

Delta method allows one to construct confidence intervals. However the method involves substantial computational difficulties, such as inverting the Fisher information matrix, which is quite large if the number of parameters is large.

The paper proposes an alternative method. it involves regularisation and implies asymptotical convergence. The method is easier computationally and converges asymptotically. A number of convergence results are proven. The method returns good performance in practice.

**Questions:**

Please correct me if I am wrong. I might have missed important points.

**Limitations:**

Yes.

**Strengths And Weaknesses:**

I am worried this paper belongs to the domain of information theory rather than generic machine learning and some key points may be lost on the NeurIPS audience as they no doubts were lost on me.

Otherwise the paper appears to make a substantial contribution.

---

> ### Author Response · Authors · 2022-08-02
> **Response to Reviewer ZZii**
>
> Thank you for the encouraging remarks and recognizing the strength of our contribution. Your summary of the paper is accurate.
>
> In terms of categorization, we do strongly believe the contribution squarely fits at NeurIPS, generally focusing on the intersection of statistics and ML and specifically providing uncertainty-quantification tools for users of ML methods. There are many prominent examples of ML work on modern statistical inference, dealing with the larger, richer, and more challenging settings commonly encountered in machine learning. To name a few examples, all of which specifically deal with modern methods for constructing frequentist confidence intervals and all of which are at NeurIPS: "Confidence Intervals and Hypothesis Testing for High-Dimensional Statistical Models" from NeurIPS 2013, "Statistical Inference for Pairwise Graphical Models Using Score Matching" from 2016, "Post-Contextual-Bandit Inference" from NeurIPS 2021, "Asymptotics of the Bootstrap via Stability with Applications to Inference with Model Selection" from NeurIPS 2021, and "Inference for Batched Bandits" from NeurIPS 2020. We hope you agree.

---

### Meta-Review · Area_Chair_NhtP · 2022-08-25

**Recommendation:** Accept
**Confidence:** Certain

**Metareview:**

The paper proposes an original new tool to access the uncertainty of a machine learning model. The authors agreed that it is a valuable contribution to our community and deserves acceptance.

Importantly, all reviewers mention that there is room for improvement, both in the presentation containing ambiguities and in the empirical evaluation that needs strengthening. The authors properly acknowledge this in their rebuttal, and a consensus emerged from the discussion that those shortcomings are fixable.

Thus, I kindly ask the authors to carefully revise their paper for the camera ready by implementing all the changes they committed to and considering all reviewer's comments. This includes (but is not limited to):
- Make the paper more accessible for a broader ML audience by including more background, clarifying the scope of the paper in the abstract and introduction, and discussing what kind of uncertainty is studied throughout the paper;
- Adding the supplemental results reported during the authors-reviewers discussion;
- Report the accuracy of the method using synthetic data in which we can simulate the actual sampling uncertainty ([asked by Reviewer  41Kr](https://openreview.net/forum?id=etY_XXnPkoC&noteId=CLiYTiN03Ed))


**Award:**

No

---

### Decision · Program_Chairs · 2022-09-14

Accept